# Efficiently Scaling LLM Reasoning with Certaindex

**Yichao Fu**[*1]   **Junda Chen**[*1]   **Siqi Zhu**[1°]   **Zheyu Fu**[1]   **Zhongdongming Dai**[1]
**Yonghao Zhuang**[2]   **Yian Ma**[1]   **Aurick Qiao**[3]   **Tajana Rosing**[1]   **Ion Stoica**[4]   **Hao Zhang**[1]
[1]UCSD   [2]Carnegie Mellon University   [3]Snowflake   [4]UC Berkeley

## Abstract

Test-time reasoning algorithms such as chain-of-thought, self-consistency, and MCTS enhance LLM problem-solving but can wastefully generate many tokens without improving accuracy. At the same time, we observe that these algorithms exhibit answer stabilization: their intermediate solutions often cease to change after a certain point, and further investment of compute does not change their final answer. To quantify this phenomenon, we introduce *Certaindex*, an algorithm-agnostic metric measuring this evolving stability, signaling when further computation is unlikely to alter the final result. Certaindex is lightweight, can accelerate reasoning program inference via early exit, and further enables dynamic token allocation, gang scheduling, and many opportunities when integrated with real-world LLM serving systems. To quantify real-world benefits, we built Certaindex as a scheduler into *Dynasor*, our reasoning-aware LLM serving system, and demonstrate up to 50% compute savings and $3.3\times$ higher throughput in real workloads with no accuracy drop. Our code is available at `https://github.com/hao-ai-lab/Dynasor.git`.

## 1 Introduction

Large language models (LLMs) have recently demonstrated remarkable ability in solving complex problems. Central to these advances are *test-time scaling algorithms* [1; 2; 3], which allocate additional inference resources to progressively enhance model accuracy. Whether the reasoning algorithm is externally programmed (e.g. Self-Consistency [4], MCTS [5; 6], etc.) or internalized in the model (Chain-of-Thought based models such as Deepseek-R1 [7]), or a combination of both (as used in OpenAI-o3 [8], Grok3 [9], Gemini 2.5 Pro [10]), these methods fundamentally empower LLMs to tackle challenging problems more effectively.

However, we observe that LLM reasoning is highly *token-inefficient*, i.e., often leads to token overuse without further accuracy gains. This phenomenon is most observable in reasoning models: models like Deepseek-R1 can generate excessive amount of tokens, resulting in substantial 3x more tokens than it actually needs (see Figure 2). Similar token overuse behavior is also observed in recent studies [11; 12]. This inefficiency arises because existing reasoning algorithms lack mechanisms to detect diminishing returns in accuracy and terminate inference early, thereby wasting resources without gains in solution quality.

To address this inefficiency, we leverage a key observation about LLM reasoning: LLMs often signal when they've "settled" on an answer during reasoning. In particular, we study reasoning models with Chain-of-Thought (CoT, §2.1): we force the model to periodically output a result in the middle of its reasoning process, and examine how these intermediate answers evolve across successive steps. We find these intermediate answers frequently stabilize; that is, they rarely change after a certain number of reasoning steps, regardless of whether the answer is ultimately correct or not. This highly indicates that the model itself has reached a point of "high certainty" that further computation is unlikely to alter the final result, justifying safe termination. Conversely, significant variance in the

---

*Equal contribution.   °Work done while interning at UCSD

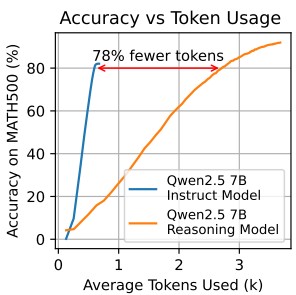

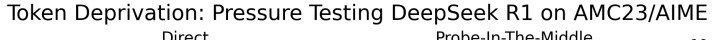

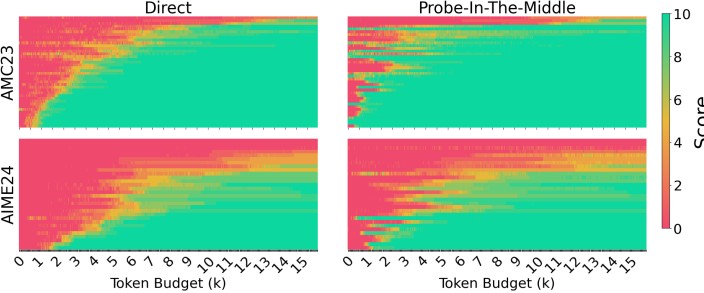

Figure 1: The token efficiency curve for the traditional model is much steeper than reasoning model.

Figure 2: DeepSeek R1 performance on AMC23 (upper) and AIME24 (lower) at varying token budgets (scoring lowest to highest over 10 attempts). (Left) Standard reasoning with late answer outputs. (Right) Early answer extraction using Probe-In-The-Middle technique.

outputs suggests the model remains uncertain and is still exploring different solution paths, warranting continued reasoning. By monitoring these signals, we can decide if we can terminate LLM reasoning early, saving computational resources without token waste.

Taking the certainty as a signal, we further explore advanced algorithms such as Self-Consistency (SC), Monte Carlo Tree Search (MCTS), and Rebase [13] – all of which have seen empirical success in improving CoT-based reasoning in practice. While these algorithms have distinct mechanisms, we found they share a fundamental behavior: as more computational resources (e.g., tokens, reasoning steps) are invested, their answers tend to stabilize. This common observation indicated that a general measure of this evolving confidence is possible. To formalize this as a unified metric, we introduce *certaindex* (§3.1) – a universal metric designed to capture this developing certainty. Certaindex generalizes certainty into a normalized confidence score that serves as a proxy to measure reasoning progress. Regardless of what the reasoning algorithm is, high certaindex values indicate that the model's answer is unlikely to change significantly with further computation.

Even better, we find that certaindex offers a narrorw interface to real-world LLM serving engines. Because certaindex is light-weight, it enables opportunities including early exit, dynamic token allocation across reasoning queries, and gang scheduling in multi-stage reasoning programs, all with almost no scheduling overhead (§3.2). To this end, we develop *Dynasor* – a reasoning-aware end-to-end serving system that leverage certaindex to optimize compute usage in both batch and online serving (§3.3). Dynasor employs certaindex to adaptively allocate token budgets to reduce the compute cost, and enables gang scheduling to increase request SLO attainment and throughput. Our evaluations on various datasets, LLMs, and reasoning algorithms show that in batch inference, it saves up to 50% compute to reach the same overall accuracy; and in online serving, it sustains up to $3.3\times$ more queries or achieves $4.7\times$ tighter latency SLOs at the same attainment rates.

In summary, this paper makes the following contributions:

1. We identify significant token overuse in concurrent reasoning models and introduce Probe-In-The-Middle, a method that extracts intermediate answers during CoT to detect convergence and enable early stopping – validated both empirically and theoretically.

2. We develop *Certaindex*, a unified metric that quantifies reasoning progress across diverse strategies (CoT, SC, MCTS, Rebase), enabling adaptive compute allocation at test time.

3. We build *Dynasor* as a reasoning-aware serving system that leverages certaindex to dynamically allocate tokens and co-schedule multi-stage reasoning steps, achieving up to 50% compute savings in batch serving and 3.3× throughput gains in online serving.

## 2 Systematic Token Overuse – A Case Study in CoT

Recent work has shown that allocating more compute at inference time, so-called "test-time scaling", consistently boosts performance on hard reasoning tasks [8; 14; 15]. Methods such as Chain-of-Thought (CoT) [16], Best-of-N sampling [17; 18], Self-Consistency (SC) [4], and search-based

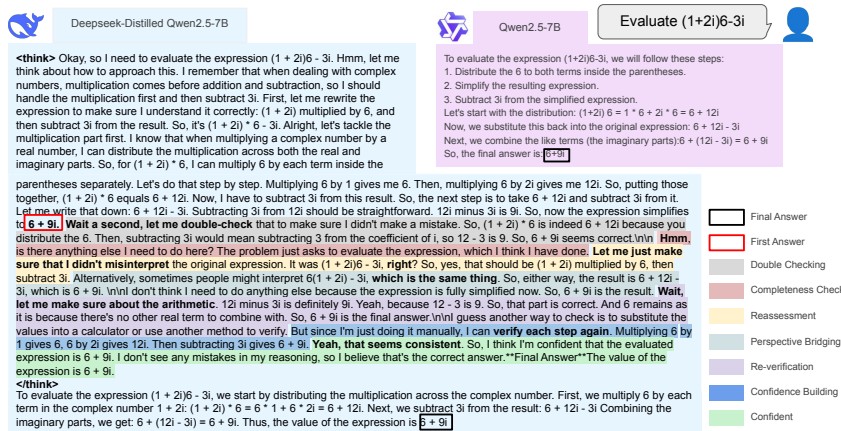

Figure 3: An Example of **Self-Doubt** Comparing a Reasoning Model (Deepseek-distilled Qwen-2.5 7B) vs. a Traditional Model (Qwen-2.5 7B) on a Problem from the MATH500 Dataset

algorithms (e.g., MCTS [5; 6], guided beam search [19], or Rebase [13]) all exploit more token usage to trade compute for accuracy.

While advanced reasoning algorithms boost accuracy on complex tasks, they often suffer from severe token inefficiency. To quantify this, we compare a Qwen2.5 7B (Non-Reasoning) [20] instruct model against a Qwen2.5 7B reasoning model [14] on the MATH500 dataset, gradually increase the maximum token budget for each question, and track the accuracy change for the dataset. Figure 1 reveals that the reasoning model, despite eventually achieving higher peak accuracy, required up to $4.5\times$ more tokens to reach the same accuracy as the instruct model.

Why do reasoning models overuse so many tokens? To answer this question, we conduct a case study focused on CoT reasoning models in math datasets (i.e., AMC23, AME24) to understand where this overuse occurs and why. We develop and use *"Probe-In-The-Middle"* (§2.1), periodically inserting a prompt such as *"Oh, I suddenly got the answer to the whole problem, Final Answer: boxed{"* to force the model to output its answer mid-reasoning, and record whether this answer is correct. Figure 2 presents the LLM's accuracy for each question at various token budget allocations with (left) and without (right) "Probe-In-The-Middle". Each row in this figure gives us an approximation of the number of tokens spent (left) vs actually needed (right) to arrive at the correct answer. For example, on AMC23, the model produces a median of **2.7K** tokens, but can typically output the correct answer by a median of **830** tokens.

We further zoom in to one case, and realize the model the major source of inefficiency stem from what we called "self-doubt". Figure 3 shows an annotated output where a reasoning model (Deepseek-distilled Qwen2.5-7B) answers a simple math question. Even as the reasoning model reached the correct answer very early ($\sim 300$ tokens), it continues the reasoning process, spending excessive tokens re-evaluating already correct answers, checking completeness, re-verifying assumptions, building confidence, etc. Several recent studies [11; 12] have also found similar phenomenon, indicating that this pattern of "self-doubt" is not an isolated issue specific to a single model or dataset but rather reflects a systematic tendency in current reasoning models.

## 2.1 Addressing Token Overuse In Long CoT Reasoning by Probing-In-The-Middle

How do we address token overuse? We conjecture that LLMs, during their reasoning, will emit measurable signals to their internal reasoning state. Indeed, much prior work indicates that LLMs might possess an internal awareness of their answer confidence, akin to 'LLMs knows when they know' [21]. If these signals can be accurately detected, we can use them to stop the reasoning process without sacrificing accuracy, thus reducing token usage.

Our high-level idea is that if an LLM consistently produces the same intermediate answer over a sequence of reasoning steps, it indicates a high degree of *certainty* or stability for that answer, whether or not that answer itself is correct or not. Once such stability is reached, additional reasoning

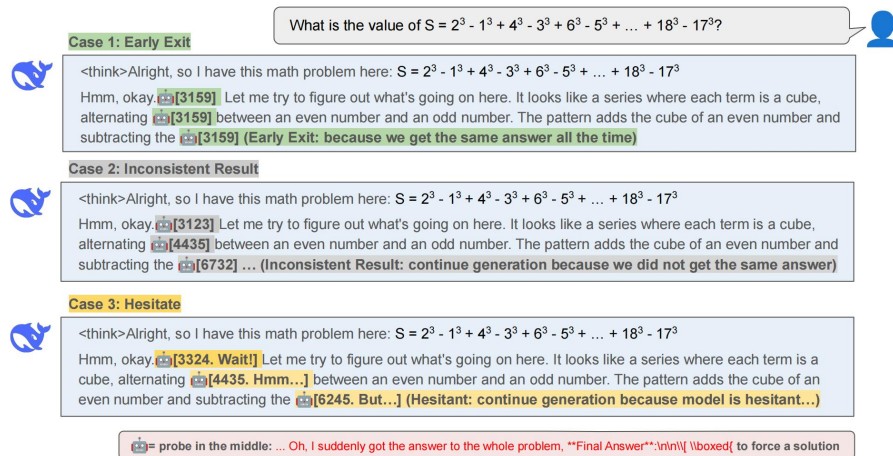

Figure 4: Illustration of Dynasor on CoT: (1) Probe-In-The-Middle for answer extraction, (2) early exit based on certainty (case 1), (3) post-generation validation for hesitation words (e.g., "wait") (case 3), and (4) continue if not certain enough (case 2)

is unlikely to alter the outcome, presenting an opportunity for early termination. Conversely, if intermediate answers remain highly varied or fail to converge after significant effort, it may signal the problem as intractable for the model, also justifying early termination. Given this insight, we use "Probe-In-The-Middle" to first capture intermediate answers (e.g. every 64 tokens) as the model reasons, and then assess the *certainty* based on the consistency of these probed answers to decide whether to stop generation. By monitoring this certainty, we can truncate the reasoning chain, thus reclaiming wasted compute without sacrificing accuracy.

**Probe-In-The-Middle.** Figure 4 shows the high-level process. To capture whether the reasoning LLM has settled on its final answer, we interleave periodic "probes" into its reasoning process. Specifically, after a set number of generated tokens, we append the prompt: *"Oh, I suddenly got the answer to the whole problem. Final Answer: boxed{"*, and record the output answer. This prompt forces the model to take a guess for the final answer at this specific point in its reasoning. Note that the exact phrasing of the extraction prompt is not critical. What matters is that it effectively guides the model to produce an answer immediately.

Concretely, we split the full reasoning chain into $m$ steps, $Y_1, Y_2, \ldots, Y_m$, where each $Y_k$ corresponds to a fixed token interval (e.g., 64 tokens). At each step $k$, we sample $Y_k \sim P_k\big(Y_k \mid x, Y_1, \ldots, Y_{k-1}\big)$. All probe tokens and their responses are then discarded before resuming the original decoding path. This lightweight probing mechanism allows us to pinpoint exactly when the model's answer converges—enabling systematic comparison of token efficiency across models on large datasets.

**Certainty Assessment via Answer Consistency.** Using the answers captured through Probe-In-The-Middle, we perform an immediate consistency check to assess the model's certainty at each probing step. Concretely, let $\{y_1, \ldots, y_m\}$ be the sequence of probed answers from step 1 to step $m$. We measure consistency over a sliding window of width $w$ steps: $C_k = \frac{1}{w} \sum_{j=k-w+1}^{k} \mathbf{I}\big[y_j = y_k\big]$, where $\mathbf{I}[\cdot]$ is the indicator function. Once $C_k \geq \tau$ for some threshold $\tau \in (0, 1]$, we deem the model sufficiently certain and terminate generation at step $k$ (case 1 in Figure 4). Otherwise, we will continue reasoning (case 2 in Figure 4) until the budget is drained or some other stopping criteria is matched.

**Post-Generation Validation.** In addition to the answer consistency, we found that some linguistics markers like "wait" or "hmm" also indicates uncertainty. If we find these uncertainty indicators in the probed answers (case 3 in Figure 4), we will mark this answer as unconfident and omit this responses from the consistency test. This validation mechanism works synergistically with the consistency assessment to create a robust certainty metric.

**Theoretical Grounding and Efficacy for CoT.** The rationale for early terminating based on this observed stability is not merely empirical. In § 3.4, we provide a theoretical analysis. This analysis demonstrates that if a reasoning chain consistently yields the same answer across several consecutive reasoning steps, then as the number of these consistent steps increases, the observed answer provably converges to the final answer the CoT would have produced with full, unconstrained generation.

Applying "Probe-In-The-Middle" to CoT reasoning on datasets like MATH500 (§4.1) demonstrates significant token efficiency gains. Identifying answer convergence allows early termination, which reduces computational overhead while maintaining improving accuracy. This success with CoT motivates *certaindex*, our universal metric extending the certainty-driven early exit concept to a broader spectrum of reasoning algorithms including Self-Consistency, MCTS, and Rebase.

# 3 Certaindex in General Reasoning Algorithms

## 3.1 Generalizing Certainty to a Broad Spectrum of Reasoning Algorithms

Beyond just CoT, a wide spectrum of LLM reasoning algorithms also exhibit (or can be equipped with) measures that reflect their progress towards a stable answer. For instance, iterative refinement methods (e.g., CoT) may show convergence in output, while search-based algorithms (e.g., MCTS) might reach a state where further exploration yields diminishing returns. Recognizing this common signal of "certainty" or solution stability across different algorithms, we propose *certaindex*—a unified, algorithm-agnostic confidence metric. High certaindex indicates close proximity to a solution or that additional computation is unlikely to improve the outcome.

While previous research has explored uncertainty estimation through various approaches (semantic metrics[22; 23], log-probability entropy [21; 24], and hidden state analysis [25; 26; 27]), certaindex offers a practical, integrative measure of model confidence suitable for direct implementation in LLM serving engines (e.g., SGLang and vllm). This facilitates a scheduling layer optimized for LLM reasoning workloads. In the following sections, we will define specific instantiations of certaindex for two common reasoning algorithm archetypes.

**Certaindex in reasoning algorithms with multiple reasoning paths.** Reasoning algorithms like Self-Consistency (SC), MCTS, and Rebase expand multiple reasoning paths to derive aggregated final answers (Appendix B). To quantify LLM's certainty among these paths, we employ semantic entropy [22]. Given a question, $n$ reasoning paths are generated and clustered into $m$ groups based on their answers: $C_1, C_2, \ldots, C_m$, where $|C_i|$ denotes the number of paths in answer group $C_i$. The semantic entropy is calculated as $\mathcal{H} = -\sum_{i=1}^{m} \frac{|C_i|}{n} \log \frac{|C_i|}{n}$. We normalize $\mathcal{H}$ by its maximum $\log n$ to obtain certaindex: $\tilde{\mathcal{H}} = \frac{\log n - \mathcal{H}}{\log n} \in [0, 1]$. For closed-form tasks (e.g., arithmetic or multiple-choice), we group outputs by exact string matching; for open-ended generation (e.g., code [28] or flexible mathematical expressions [29]), we compute pairwise similarities with a small embedding model [30] and cluster. Both approaches incur little to no overhead compared to LLM inference.

**Certaindex in reasoning with a reward model.** For reasoning algorithms that incorporate a reward model (e.g., MCTS, Rebase), we simply use the reward model's normalized output $\mathcal{R} \in [0, 1]$ as a measure of certainty. This approach builds on prior research demonstrating that reward signals can effectively guide resource allocation in program execution [31]. We collect the terminal reward scores from each reasoning path and aggregate them to compute certaindex: for MCTS, we take the average reward; for Rebase, the maximum reward. A higher aggregated reward indicates stronger certainty in the reasoning paths' validity. These reward scores are obtained during normal execution and therefore incur no extra overhead during LLM inference.

## 3.2 Effectiveness of Certaindex for Dynamic Token Budgeting in Reasoning Optimization

Since certaindex tracks the model's self-assessed proximity to a final solution, it can effectively guide token budgeting to prevent unnecessary token expenditure. In this section, we empirically validate this core hypothesis: that higher certaindex values consistently predict fewer remaining token needs to reach a correct solution. This relationship holds across various models, reasoning algorithms, and datasets, forming the basis for smart token allocation.

**Correlation between Certaindex and Remaining Computational Effort.** Our hypothesis is that as a model becomes more "certain" (as measured by certaindex), it should be closer to outputting its final, stable answer. To test this, we compare these certaindex values against the "oracle" number of additional tokens the model actually required from that point to arrive at the correct final answer (for solvable queries). This analysis spanned 4 different model–algorithm–task combinations, with representative examples of these correlations shown in Figure 5 (further examples are in Appendix C). Across all combinations, for solvable queries, we found Pearson correlation coefficients ranging from 0.17 to 0.75, with a strong mean of 0.52. This consistently positive correlation indicates that a higher certaindex value is indeed a reliable predictor of solution proximity, requiring fewer additional tokens.

**Thresholding-based allocation.** An immediate application of certaindex for token budgeting is a thresholding-based allocation policy. If a query reaches a high certaindex value early in its reasoning, it suggests the model is confident in an answer. We select a certaindex cutoff threshold at a chosen reasoning step (e.g., after a fixed number of reasoning steps, visualized as the horizontal orange line in Figure 5). Any query whose certaindex surpasses this threshold at this step is immediately halted. This approach has two benefits: (1) it efficiently reduces token usage on straightforward cases that quickly achieve high certainty, and (2) it halts unproductive paths early, preventing wasted resources on unsolvable or confidently incorrect answers. As our empirical results show, this approach often achieves little to no drop in accuracy on the broader set of solvable queries that do not cross this early exit threshold. Thresholding-based allocation is not strictly optimal, as we show in the pareto-frontier optimal strategy in Appendix C and Figure 12. Nevertheless, thresholding-based allocation is simple and efficient in practice, making it a compelling choice for real-world deployments.

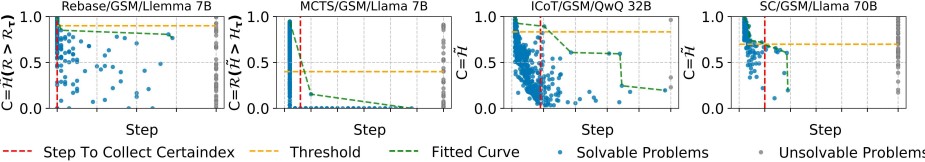

Figure 5: Correlations between *certaindex strength* (y-axis) and ground truth *steps to solution* (x-axis) with (algorithm, LLM) settings where algorithm $\in$ {SC, Rebase, MCTS, CoT} with LLM $\in$ {Llama [32] and QWQ [33]} on GSM8K. Certaindex is measured at the reasoning step marked by the red line. The orange line shows the thresholding-based allocation. The green line shows a more fine-grained approach through curve fitting. All plots except MCTS have both certaindex values and oracle steps averaged across multiple runs to combat randomness. See Figure 12 for the full result.

## 3.3 Dynasor: A Certaindex-Driven Adaptive Compute Scheduler

The previous section demonstrated how Certaindex effectively quantifies certainty during LLM reasoning, enabling dynamic token allocation and reducing waste. Building on this, we introduce *Dynasor* – a reasoning-aware, end-to-end serving system that leverages Certaindex for adaptive scheduling. In real-world deployments for LLM reasoning, a single user query often spawns multiple related sub-queries, which we refer to as "*reasoning programs*". These programs consist of logically connected requests that share context and computation. Traditional schedulers treat these sub-queries independently, missing opportunities for optimization. Dynasor addresses this by introducing a lightweight, Certaindex-driven scheduler that optimally manages both individual requests and entire reasoning programs. Beyond simple early exits, Dynasor implements two other key mechanisms to accelerate inference:

*Certaindex-Driven Adaptive Compute Allocation.* Dynasor dynamically adjusts token budgets for individual requests based on their real-time Certaindex values. As validated in §3.2, a high Certaindex indicates that the answer has stabilized, allowing the scheduler to reduce the token budget and save resources. This fine-grained control minimizes waste on queries that have converged or are unproductive, enhancing overall efficiency.

*Gang Scheduling.* Reasoning programs usually generate multiple related requests in parallel or at differet stages. Gang scheduling batches or prioritizes these requests together to maximize shared computation (e.g., KV-cache utilization) or early-exit high-certaindex programs to free resources.

This mechanism reduces latency and improves throughput by aligning resource allocation with real-time reasoning progress.

**Implementation.** Dynasor is implemented as a lightweight scheduling component in SGLang [34]. For each decoding step, Dynasor monitor certaindex to reallocate resources or terminate requests, and use gang scheduling to prioritize requests within a reasoning program. Dynasor only introduces $\sim 500$ lines of code into the core system, and requires no change to the model weights, reasoning algorithms, or the core execution backend. It acts as a thin layer around the standard decoding loop, yielding substantial gains in latency, throughput, and overall token efficiency. We describe the detailed design and implementation of Dynasor in Appendix E.

### 3.4 Theoretical Basis for Early Exiting in CoT without Accuracy Loss

This section gives a sketch of the theoretical foundation for our early exiting strategy based on "Probe-In-The-Middle". We show how it can terminate CoT reasoning without accuracy degradation, and we believe this theoretical grounding can be generalized to other reasoning algorithms. See Appendix F for a riguous proof.

**Sketch of Proof.** Our method assumes that the next-token distributions $P_t(Y_{t+1}|x, Y_{1..t})$ in a reasoning chain eventually converge to a stationary distribution $P_*(Y|x)$. Once $P_t = P_*$, further computational steps are redundant. The "Probe-In-The-Middle" technique empirically detects this convergence. If observed answers (and thus their empirical distributions $\hat{P}$) stabilize across several probing steps, it indicates the underlying true distributions $P_t$ (and their mixtures $\bar{P}$) have likely converged. To analyze this, we define mixture distributions:

**Definition 1** (Mixture Distributions). *Consider $n$ samples, each corresponds to $P_t, \ldots, P_{t+n}$. Denote the mixture distribution of samples $i + 1, \ldots, i + k \in [1, \ldots, n]$ to be $\bar{P}_i^{i+k} = \frac{1}{k} \sum_{j=1}^{k} P_{i+j}$.*

Our formal justification shows that if our empirical stopping criterion is met, the empirical mixture distributions $\hat{P}$ are $\epsilon/3$-close (in Total Variation distance) to the true mixture distributions $\bar{P}$. This requires a sufficient number of probes $k$, as established by the following concentration bound:

**Lemma 1.** *If $k = \widetilde{\Omega}\left(\frac{M + \log(1/\delta)}{\epsilon^2}\right)$, then $\mathrm{TV}(\bar{P}_l^{l+t}, \hat{P}_l^{l+t}) \leq \epsilon/3$, for all $l = i + 1, \ldots, i + k$, and $t = k - 1, k$, with $1 - \delta$ probability, assuming we have $M$ disjoint groups of the outputs.*

The empirical stopping criterion is triggered when the TV distance between consecutively estimated mixture distributions, $\hat{P}_i^{i+k}$ and $\hat{P}_{i+j}^{i+j+k}$ (and similarly for mixtures of size $k - 1$), remains below a small threshold $\epsilon' = \epsilon/3$ for all $1 \leq j \leq k$ (or $1 \leq j \leq k - 1$ respectively). When this empirical stability is observed, and $k$ is sufficiently large as per Lemma 1 (ensuring $\mathrm{TV}(\bar{P}, \hat{P}) \leq \epsilon/3$), the triangle inequality implies that the true underlying mixture distributions $\bar{P}_i^{i+k}$ and $\bar{P}_{i+j}^{i+j+k}$ are also $\epsilon$-close to each other (i.e., $\mathrm{TV}(\bar{P}_i^{i+k}, \bar{P}_{i+j}^{i+j+k}) \leq \epsilon$).

The connection between the stability of these true mixture distributions and the stability of individual distributions $P_t$ (which ultimately implies convergence to $P_*$ given Assumption 1) is provided by:

**Lemma 2.** *If $\mathrm{TV}(\bar{P}_i^{i+k}, \bar{P}_{i+j}^{i+j+k}) = 0$, for any $1 \leq j \leq k$, and if $\mathrm{TV}(\bar{P}_i^{i+k-1}, \bar{P}_{i+j}^{i+j+k-1}) = 0$, for any $1 \leq j \leq k - 1$, then Eq. (2) holds for any $t \geq i$, and $T < 2k - 1$.*

Therefore, observing empirical stability with a sufficiently large $k$ indicates that the true individual distributions $P_t$ have stabilized and are $\epsilon$-close to the stationary distribution $P_*(Y|x)$. This justifies that early termination preserves accuracy up to a tolerance $\epsilon$. The effective $k$ must satisfy both the concentration requirement (Lemma 1) and be large enough relative to $k^*$ from Assumption 1.

## 4 Evaluation

To show the effectiveness of certaindex in real-world workloads, we evaluate certaindex and the end-to-end serving system Dynasor in CoT [16] and three other reasoning algorithms (SC [4], Rebase [13], MCTS [6; 5]) on diverse datasets [35; 36; 37; 28; 20; 38]. For brevity, we provide detailed experimental setup in Appendix G.

Our experiments try to answer two main questions: (1) **Batch workload**: How effectively does certaindex reduce token consumption while preserving accuracy in real-world batch inference work-

loads? (§4.1); (2) **Online workload**: To what extent does Dynasor improve end-to-end performance in online serving scenarios? (§4.2). In addition, we perform ablation studies in (§4.3) to compare certaindex against other signals and selection of certaindex thresholds at runtime. We also perform a few more Dynasor system ablation studies in Appendix H.

## 4.1 Batch Inference with Certaindex

To evaluate the effectiveness of certaindex in batch inference, we evaluate Dynasor on CoT with reasoning models, and compare the tokens-accuracy tradeoff curve for different setups. We also evaluate certaindex on other reasoning programs (SC, MCTS, Rebase) to see if the effect generalizes. See detailed experiment setup in Appendix G.1 for CoT, and Appendix G.2 for SC, MCTS and Rebase.

**CoT.** Figure 6 shows our approach achieve significant token savings across all configurations, reducing token usage by 11-29% while maintaining same accuracy as the baseline. For the 10% of problems where our method achieves the highest token reduction, we observe savings of 34% on AIME and 53% on MATH500. In particular, for the top 1% of problems, we achieve even more substantial reductions of 53% on AIME and 81% on MATH500. A similar result is shown in Deepseek-R1 (See Appendix H.1). The primary gains of the token savings comes from the effective probing mechanism to identify self-doubt in CoT reasoning, and using certaindex to early-stopping.

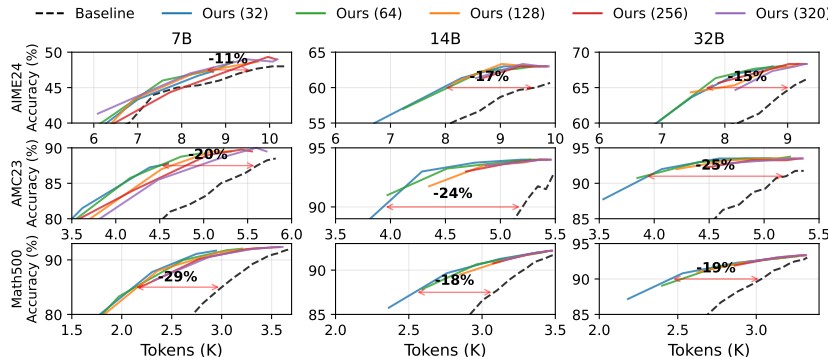

Figure 6: Comparing Dynasor Token-Accuracy Curve Across Deepseek Qwen-Distilled Model Scales (7B, 14B, 32B) and Datasets (AIME24, AMC23, Math500)

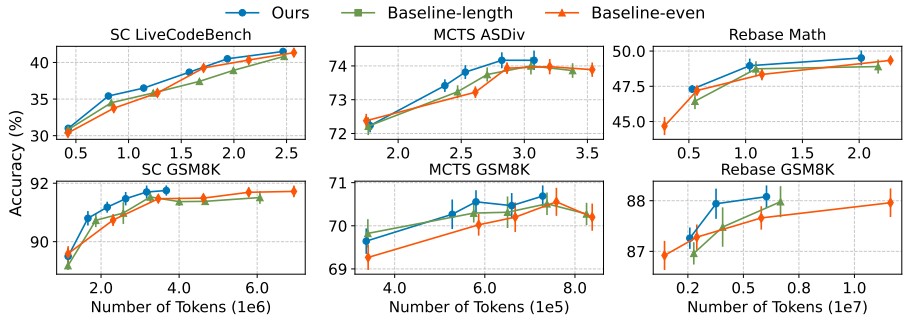

Figure 7: Token-to-accuracy metric on batch processing workloads. Mean performance and std (error bars) of 10 runs are reported. Baseline-even allocates resource uniformly across all reasoning program. Baseline-length uses Detect@knob (Table 1) as the program's process signal.

**SC, MCTS, Rebase**. To show our method generalizes to other reasoning algorithms, we also compare Dynasor against a modified SGLang intra-program scheduler using the following policies (1) **baseline-even**, which allocates resources uniformly; and (2) **baseline-length** which uses Detect@knob, the cumulative tokens generated at a specific step (Table 3) as the proxy.

Figure 7 show certaindex consistently reduces token usage by 9–52% across workloads compared to both baselines, without loss in accuracy. Notably, we achieve over 47% savings on SC-GSM8K and

over 50% on Rebase-Math, highlighting Dynasor's efficiency across diverse benchmarks. These gains primarily come from the intra-program scheduling algorithm(§ E.2.1), which accurately identifies high-certaindex programs and terminates them early without accuracy loss. This early termination strategy significantly reduces resource consumption by eliminating unnecessary sampling compared with baseline-even. In contrast, baseline-length leads to accuracy degradation even with less aggressive compute pruning, highlighting the effectiveness of certaindex-based resource allocation.

## 4.2 Online Serving with Certaindex

We evaluate Dynasor against SGLang [34] and Parrot [39] with different inter-program schedulers. We use P90 deadline attainment as the SLO attainment (detailed setup in Appendix G.3). Each row in Figure 8 presents a key performance trade-offs in online serving: (a) Program arrival rate vs SLO attainment: what percentage of programs can meet the SLO as the arrival rate increases; (b) SLO scale vs SLO attainment: how tightly a service provider can configure the SLO while maintaining reliable attainment. (c) Tradeoff between Accuracy and SLO attainment.

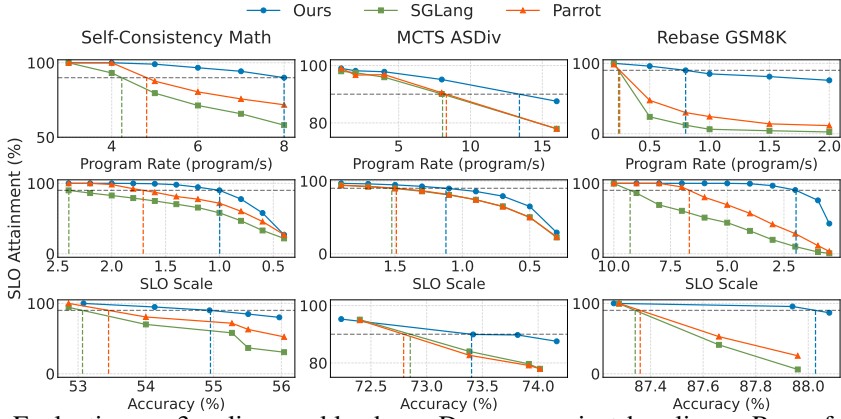

Figure 8: Evaluation on 3 online workloads on Dynasor against baselines. Rows from top to bottom: (a) Program rate vs SLO attainment, (b) SLO scale vs SLO attainment, (c) Accuracy vs SLO Attainment.

**(a) Rate vs. SLO attainment**. Figure 8(a) shows that Dynasor achieves much higher sustainable request rates under P90 deadline attainment: $1.6 - 3.3\times$ and $1.6 - 3.2\times$ compared to SGLang and Parrot, resp. These gains stem from two key factors. First, our intra-program scheduler identifies and early-terminates programs with high confidence, freeing resources for pending requests. Second, our inter-program scheduler prioritizes requests from the same program, increasing turnover rate.

**(b) SLO scale vs. SLO attainment**. Figure 8(b) shows deadline attainment under a fixed request rate for different systems. At these rates, Dynasor allows tighter SLO scales: $1.3 - 4.7\times$ tighter than SGLang, and $1.7 - 3.3\times$ tighter than Parrot. The source of gain is similar to (a): intra-program scheduler early-terminates requests and increase turnover rate, both increases effective request rate.

**(c) Accuracy vs SLO attainment**. Figure 8 (c) (details in Appendix G.3) shows Dynasor achieving 0.7% - 2% higher accuracy than SGLang and Parrot across all three workloads at the same SLO attainment. This improvement results from Dynasor's ability to redistribute compute between simple and hard queries, enabling it to solve more queries while maintaining SLOs that baselines could only match with significantly more compute.

**Throughput**. Our system shows equal average throughput (token per second) in all online settings compared to the baselines. The is because that under the given request rates, all workloads saturate the GPU memory and making the system memory bound. This also validates the introduced scheduler has no overhead.

## 4.3 Ablation Studies

**Choosing Different Thresholds.** We show that certaindex threshold selection is vital. Fig. 9 demonstrates the impact of different certaindex thresholds. For SC on GSM8k, we select an entropy

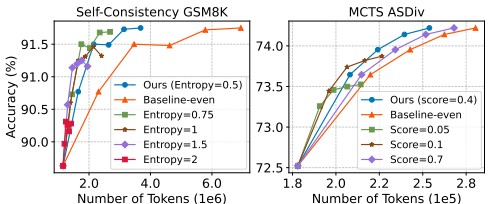

Figure 9: Performance comparison with different entropy threshold or reward score threshold.

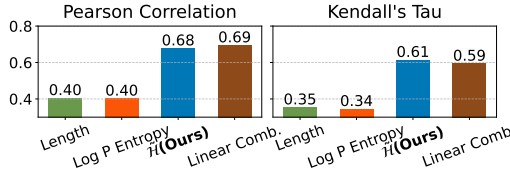

Figure 10: Correlation between certainty and mean steps required on solvable problems.

threshold of 0.5, as higher thresholds led to overly aggressive termination and accuracy degradation. Similarly, for MCTS on ASDiv, we choose a score threshold of 0.4, as lower thresholds (<0.4) decreased accuracy while higher thresholds (>0.4) increased token consumption without proportional accuracy gains. These results highlight the importance of careful threshold selection in balancing compute efficiency and accuracy. In practice, Dynasor uses a profiler-guided approach to select these thresholds (see detailed discussion in Appendix E.2.2)

**Choosing Different signals.** We compare certaindex with two alternative metrics for estimating resource needs. (1) *Reasoning path length*: Longer reasoning paths (more tokens) often indicate harder problems, suggesting a potential correlation between path length and compute needs (e.g., more reasoning paths). (2) *Mean normalized log probability*: Established as a measure of LLM confidence [24], higher log probabilities may correlate fewer samples needed for correct answers in SC. Using the (SC, GSM8K, Llama3.1-8B-instruct) setup, we evaluated certaindex alongside these metrics. Four metrics were tested: certaindex's entropy measure $\mathcal{H}$, mean output length [40], mean normalized log probability [24], and their linear combinations.

As shown in Fig. 10, $\mathcal{H}$ achieved the strongest correlation with ground truth compute requirements (Pearson Correlation of 0.68, Kendall's Tau of 0.61), outperforming other metrics and matching complex combinations. These results confirm that certaindex is a simple yet effective proxy for estimating inference computational demands, offering robust performance across tasks and models. Our end-to-end *token-to-accuracy* evaluations in Appendix H compare different signals for resource allocation, confirming certaindex's superior performance.

**Other ablations.** We also ablate the contribution to scheduling algorithm (§ H.2), fairness analysis (§ H.3), and comparing static thresholding vs fine-grained resource allocation (§ H.4).

## 5   Limitation and Broader Societal Impact Discussion

**Limitation.** Our study primarily focused on optimizing token allocation through certaindex, but did not explore its integration with advanced serving techniques like PD disaggregation or chunked prefill. The potential impact of these integrations on the latency-accuracy trade-off in real-world workloads remains an open area for future research.

**Broader Impact.** Improvements of Dynasor in LLM inference efficiency can further democratize access to powerful models. By reducing computational costs and energy consumption without model changes, our method enables more sustainable and scalable LLM serving. However, we acknowledge potential risks, including bias in early exits, security vulnerabilities through side-channel attack for certaindex in multi-tenant settings, and manipulation risks if malicious inputs exploit the mechanism.

## 6   Conclusion

In this paper, we show that LLM reasoning algorithms frequently exhibit answer stabilization, leading to unnecessary token generation. We developed certaindex, an algorithm-agnostic metric, to detect this stability and enable early exit from the reasoning process. When we implement Certaindex as a scheduler into Dynasor, our reasoning-aware serving system, this simple yet effective mechanism results in significant practical benefits, including up to 50% compute savings and 3.3x higher throughput in production systems, without accuracy drop. Certaindex thus represents a key advancement in making sophisticated LLM reasoning more efficient and scalable.

## Acknowledgment

We sincerely thank the anonymous reviewers for their valuable feedback. We gratefully acknowledge NVIDIA for providing computational resources for this project, and Snowflake for their contributions and support. This work was supported in part by PRISM, and by the National Science Foundation (NSF) under grants 2112665, 2112167, 2003279, 2120019, 2211386, 2052809, and 1911095.

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

# A Related Works

**LLM Reasoning with Test-Time Scaling.** Large language models (LLMs) are increasingly being augmented with explicit step-by-step reasoning programs to tackle complex tasks. Chain-of-Thought (CoT) prompting [16] elicits step-by-step natural-language rationales, raising task accuracy with simple trigger phrases. Self-Consistency [4] enhances CoT by sampling multiple reasoning paths and selecting the most frequent answer, while Best-of-N sampling [17; 18] generates multiple candidates and selects the highest-scoring solution. Tree-of-Thoughts (ToT)[41] casts reasoning as tree search, where nodes represent intermediate states and search algorithms explore promising branches, enabling backtracking and global evaluation. The ReAct[42] framework structures timesteps as (*Thought, Action, Observation*) triplets, allowing LLMs to reason internally and invoke external tools dynamically. More sophisticated approaches leverage Monte Carlo Tree Search (MCTS)[5; 6], guided beam search[43], and Rebase [13] to systematically explore reasoning spaces. More recent work has established Test-Time-Scaling as a key paradigm for eliciting deeper reasoning from fixed-parameter LLMs without retraining [3]. The s1 framework introduces budget-forcing rules by appending "Wait" tokens or terminating early to dynamically manage compute allocation [1]. Further work shows that compute-optimal allocation favors parallel sampling over sequential deepening [44], that methods like SETS interleave sampling with self-verification [45], and that advances in meta-generation algorithms [46] and generative reward models [47] can further enhance inference-time scaling. These methods exploit multi-step reasoning, representing a fundamental shift toward inference-time scaling in modern LLM applications. However, the gains in accuracy often come at the cost of efficiency: multi-step reasoning produces far more tokens, leading to higher computational cost and latency. This trade-off sets the stage for Dynasor, which is a system designed to address these efficiency challenges by intelligently managing token usage and early exits.

**Efficient CoT Reasoning.** Efficient LLM CoT reasoning targets reducing excessive token usage in long reasoning chains [48]. Model merging blends fast shallow and thorough deep reasoning models for CoT inference [49; 50]. Mid-generation self-evaluation prunes unpromising continuations on the fly [51]. Specialized fine-tuning compresses rationales by skipping irrelevant tokens [52; 53; 54; 55; 12; 11], while reinforcement-based methods learn to trim overlong thought sequences [12]. Parameter-space controls such as CoT-Valve adjust chain length based on learned cues [56]. Token-budget-aware frameworks estimate question difficulty to allocate computation dynamically [56], and verification-guided systems like FlashThink identify when reasoning can safely terminate [57]. While each approach improves the trade-off between compute cost and reasoning quality, most depend on weight modifications, additional modules, or extensive retraining, complicating real-world deployment. By contrast, Dynasor adds only a lightweight proxy scheduler that manages reasoning depth within standard inference pipelines, guided by *certaindex* for efficient reasoning and consistent with observations that CoT reasoning often settles early [57].

**Certainty-Driven Early Stopping for Efficient LLM Reasoning.** Recent work has focused on cutting reasoning costs by leveraging model certainty signals. Adaptive-Consistency [58] introduces a lightweight, model-agnostic stopping rule. Early-Stopping Self-Consistency [59] reduces sampling cost by halting once answer distributions converge, while Reasoning-Aware Self-Consistency [60] adjusts the number of samples required based on the quality and agreement of reasoning paths. Concurrent work Dynamic Early Exit [61] detects high-confidence points during Chain-of-Thought reasoning to stop early. Self-Consistency Preference Optimization [62] improves reasoning by training the model to favor consistent answers. ConCISE [63] shortens reasoning steps by reinforcing token-level confidence and reducing unnecessary reflections. An information-theoretic method [64] introduces two metrics to measure how far the model's reasoning is from ideal and how much each step contributes, stopping once confidence (measured via entropy) is high enough. Certainty-based adaptive routing [65] and program-aided reasoning [66] confirm that model confidence often correlates strongly with correctness. Another method [67] checks hidden states to predict whether the answer is correct. However, these methods focus primarily on single reasoning paradigms (self-consistency or CoT), neglecting other reasoning algorithms and parallelism opportunities. In contrast, Certaindex provides a theoretically grounded stopping criterion that generalizes beyond self-consistency or CoT to multiple reasoning programs, including CoT with majority voting, MCTS, and reward-guided tree search, while adding only a lightweight proxy scheduler that preserves both simplicity and deployment compatibility.

**Tool-Validated Reasoning.** Several approaches have been proposed to enhance LLM generation by injecting external validation via test execution or unit-test generation to improve answer quality. CodeT [68] automatically generates unit tests with the same LLM, then applies dual execution consensus among code candidates to select the most reliable solution. Self-Debugging [69] prompts the model to detect when generated code fails, let it rubber-duck the error, and autonomously patch the program based on runtime feedback. LEVER [70] trains a verifier that jointly considers the input task, code, and its execution outcomes to rerank and filter generated programs more accurately. DOCE [71] combines sample reranking, execution-based scoring, minimum-Bayes-risk decoding, and self-debugging into a single pipeline for execution-aware decoding optimization. While these methods rely on external tools, such as runtime environments, test runners, or auxiliary verifiers. Dynasor takes a different path: it only leverages the LLM itself, harnessing internal uncertainty and cross-consistency signals to drive early stopping, without executing any code or requiring additional tooling.

**LLM Inference and Serving Systems.** Existing LLM serving systems improve throughput and latency via batching [72; 73; 74; 75], memory paging [76], and disaggregation [77; 78], and many other techniques. Recent frameworks such as ParrotServe [39] and SGLang [34] co-design frontend and backend for multi-query workflows and KV-cache reuse, substantially enhancing efficiency. However, these systems treat requests as independent and do not target reasoning algorithms, which exhibit dynamic scaling and within-request branching. Dynasor is the first system to exploit adaptive compute–accuracy trade-offs in reasoning workloads through a *Certaindex*-guided scheduler that dynamically allocates compute based on query difficulty.

## B  Example of LLM Reasoning Algorithms

We describe four widely used reasoning algorithms. While their specifics differ, they share two core operations: (1) *expansion*: generating tokens to expand solution trajectories, and (2) *aggregation*: combining results from trajectories to derive a final answer. Increasing compute for expansion generally improves the quality of answers during aggregation.

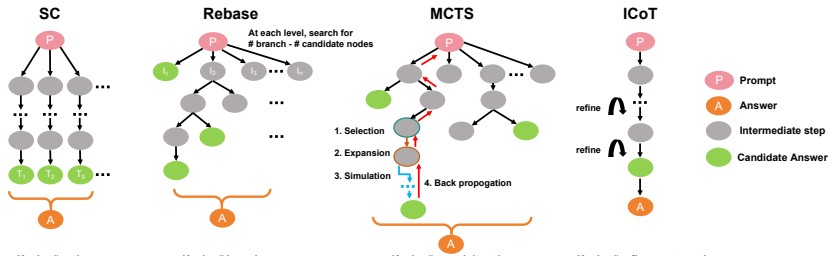

Figure 11: Illustration of the workflow of different LLM reasoning algorithms discussed in Appendix B

**Self-consistency (SC).** Fig. 11(a) depicts the computational process of SC [4]. Starting with a prompt $P$, SC expands $n$ trajectories $T_1, T_2, \ldots, T_n$ by sampling different outputs from the LLM (e.g., varying random seeds or temperature). Each trajectory represents a reasoning path that terminates in a candidate answer. During aggregation, SC applies majority voting across answers in $T_{1:n}$ and selects the one that appears most frequently. The key compute-control parameter is $n$, which dictates the number of generated trajectories.

**Rebase.** Rebase [13], as shown in Fig. 11(b), also begins by generating $n$ independent outputs $I_1, I_2, \ldots, I_n$ from the input prompt $P$. Unlike SC, these outputs represent intermediate steps, which may or may not contain the final solution. They are ranked and assigned with scores $s_1, s_2, \ldots, s_n$, typically by a learned reward model. Rebase selects nodes with higher normalized exponential scores $e^{s_i} / \sum_{j=1}^{n} e^{s_j}$ as parent nodes, from which the next $n$ reasoning steps are branched out.

This process is repeated until $n$ candidate solutions are returned. The final result is aggregated by (weighted) majority voting across all $n$ candidate answers or selecting the top-scored answer (a.k.a. *best-of-n*). Rebase structures the reasoning process as a solution tree, where tree's depth (i.e., solution

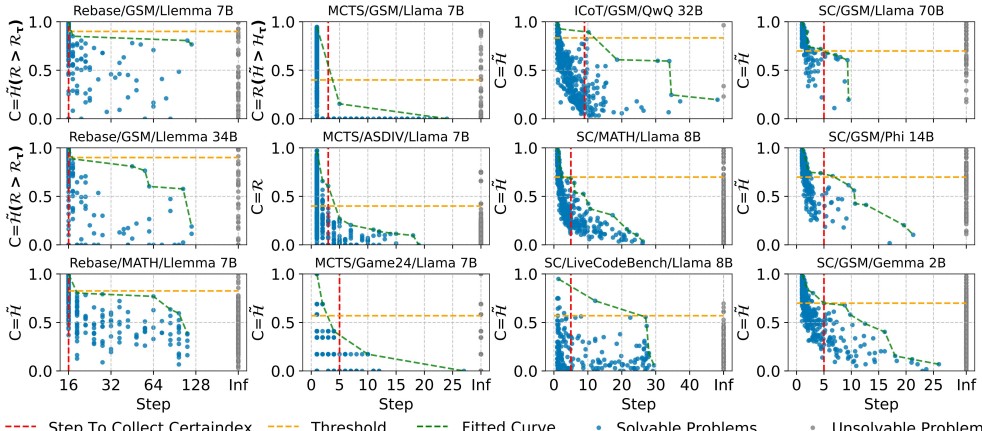

Figure 12: Correlations between *certaindex strength* (y-axis) and ground truth *steps to solution* (x-axis) on 12 (algorithm, task dataset, LLM) settings where algorithm ∈ {SC, Rebase, MCTS, ICoT}, dataset ∈ {LiveCodeBench [28], GSM8K, ASDiv [37], GAME24 [29]}, and LLM ∈ {Llama [32], Gemma [79], Phi [80], QWQ [33]}. How certaindex is measured in each setting is shown in the $y$ label of each plot. Certaindex is measured at the reasoning step marked by the red line. The orange line indicates the thresholding-based allocation. The green line illustrates a more fine-grained approach through curve fitting. For all plots (except MCTS), both certaindex values and oracle steps were averaged across multiple runs to combat randomness.

steps to reach an answer) depends on LLM outputs and its width is exactly $n$, which controls its inference-time compute.

**Monte Carlo Tree Search (MCTS).** Starting from the prompt $P$, MCTS [6; 5] iteratively builds a solution tree (Fig. 11(c)). It expands nodes by sampling continuations from the LLM step-by-step, until reaching a leaf node containing a candidate solution. Each solution is scored via a reward model or simulation, and the score is back-propagated to its ancestral nodes to update their reward values. This completes one iteration (or *rollout*). MCTS iteratively performs $n$ iterations, starting each from a node with the currently highest reward. $n$ is the key knob that controls the inference-time compute – increasing $n$ allows deeper and broader exploration of potential solutions.

**Internalized Chain-of-thought (ICoT).** Lastest LLMs, such as OpenAI o1 [15] and Qwen-QWQ [33], can internalize reasoning behaviors at training, without needing for explicit reasoning algorithms. These models generate extended chain-of-thoughts [16] sequences to derive candidate answers to reasoning queries. They iteratively refine these answers by reflecting on prior outputs and continuing sequential decoding until termination, yielding the final solution. The number of refinement rounds $n$, which corresponds to total decoded tokens, directly determines inference-time compute.

## C    Certaindex Based Resource Allocation

Large language models (LLMs) have an inherent ability to "know when they know", meaning they can self-assess their confidence in their answers. This ability, discovered in prior work such as [21], allows LLMs to indicate their certainty while generating tokens. In this section, We introduce certaindex, a measure of this confidence, as a proxy for reasoning progress: high certainty suggests the LLM is nearing a final answer or directly indicates a lower absolute compute needed to reach a final answer, whether correct or not. Our goal is to measure and track certaindex during inference, enabling dynamic resource allocation, prioritization of complex queries, scaling back simpler queries, and terminating unpromising queries. Some of the contents will overlap with § 3.1, as we want to provide a more complete narrative of the certaindex definition.

Various methods have been proposed to estimate LLM uncertainty, including semantic measures [22; 23], log probability entropy [21; 24], and hidden state-based indicators [25; 26; 27]. While

certaindex's mathematical formulation varies across reasoning algorithms, its core interpretation remains the same: the LLM's certainty in its reasoning paths.

## C.1 Measuring Certaindex

This section presents two basic formulations of certaindex.

**Certaindex in typical reasoning algorithms.** Reasoning algorithms like SC, MCTS, and Rebase expand multiple reasoning paths to derive aggregated final answers (Appendix B). To quantify LLM's certainty among these paths, we employ semantic entropy [22], which is derived from empirical entroy. Given a question $P$, $n$ reasoning paths are generated and clustered into $m$ groups based on their answers: $C_1, C_2, ..., C_m$, where $|C_i|$ denotes the number of paths in answer group $C_i$. The empirical entropy is calculated as: $\mathcal{H} = -\sum_{i=1}^{m} \frac{|C_i|}{n} \log \frac{|C_i|}{n}$.

$\mathcal{H}$ intuitively measures the diversity of answer distributions. A higher $\mathcal{H}$ indicates greater uncertainty, where reasoning paths diverge into many different answers. Conversely, a lower $\mathcal{H}$ suggests higher certainty, typically when most paths converge to the same answer (large $|C_i|$ for some group $i$). The maximum entropy $\max(\mathcal{H}) = \log n$ occurs when each path yields a unique answer ($m = n, |C_i| = 1$ for all $i$). We normalize $\mathcal{H}$ by $\max(\mathcal{H})$ to obtain certaindex:

$$\tilde{\mathcal{H}} = \frac{\max(\mathcal{H}) - \mathcal{H}}{\max(\mathcal{H})} \in [0, 1]. \tag{1}$$

For SC/Rebase/MCTS on exact-answer tasks such as arithmetic and multiple-choice, those exact answers can be extracted by applying string-matching on the generated outputs, and grouped based on their equality. In more open-ended generation tasks such as code (e.g., LiveCodeBench [28]) or flexible mathematical expressions [29], answer extraction becomes non-trivial. We employ small embedding models [30] (e.g., 100M parameters) to compute textual similarities between final outputs, and cluster reasoning paths based on semantic proximity, which enables efficient answer grouping across varying problem formats.

Both clustering methods are computationally lightweight: string matching is fast and efficient; running the embedding model and calculating similarity remains computationally insignificant compared to LLM prefill and decode operations.

**Certaindex in reasoning algorithms with a reward system.** For reasoning algorithms that incorporate a reward model (e.g., MCTS, Rebase), we simply use the reward model's normalized output $\mathcal{R} \in [0, 1]$ as a measure of certainty. This approach builds on prior research demonstrating that reward signals can effectively guide resource allocation in program execution [31]. We collect the terminal reward scores from each reasoning path and aggregate them to compute certaindex. The aggregation method varies by algorithm: for MCTS, we use the mean reward across its different paths, while for Rebase, we take the maximum reward value. A higher aggregated reward indicates stronger certainty in the reasoning paths' validity, while a lower score suggests uncertainty. These reward scores are collected during the normal execution of the reasoning algorithms and thus do not incur additional overhead.

**Combining multiple certaindex indicators.** Certaindex can also be composed by multiple signals, provided they effectively capture *certainty* during the LLM reasoning process. When multiple signals are available, they can be combined by applying individual thresholds to each metric. For instance, in our MCTS/GSM8K experiments (Fig. 12), we use two distinct metrics to measure certaindex: the reward score $\mathcal{R}$ and the entropy-based measurement $\tilde{\mathcal{H}}$. Each metric is compared against its respective threshold, $\mathcal{R}_\tau$ and $\tilde{\mathcal{H}}_\tau$ respectively. A program is considered to meet the certainty requirement only if all metrics exceed their thresholds.

## C.2 Effectiveness of Certaindex

This section empirically demonstrates that certaindex correlates strongly with the computational resources required to reach correct solutions across diverse models, reasoning algorithms, and task datasets. Higher certaindex values consistently indicate lower total compute needs.

**Correlation.** We measure certaindex at intermediate reasoning steps and analyze its correlation with the ground-truth number of steps required to yield correct answers, as determined by an oracle. Queries are categorized as solvable or unsolvable, where unsolvable queries cannot produce correct answers even with near-infinite compute. Figure 12 illustrates this correlation across 12 (model, algorithm, task) settings. On solvable queries, Pearson Correlation values between certaindex and required compute range from 0.17 to 0.75 (mean 0.52), indicating a strong correlation between high certaindex and fewer steps needed to solve a query. We next demonstrate two potential resource allocation strategies using certaindex.

**Thresholding-based allocation.** We measure certaindex at a specific reasoning step, shown as a red vertical line in each plot of Fig. 12. A straightforward allocation strategy sets a threshold value $t$ (orange horizontal lines) and halts inference for any query with certaindex exceeding $t$. In all plots, nearly no solvable queries fall in the upper-right area beyond the red and orange lines, indicating that an appropriately chosen $t$ can effectively early-terminate queries once their certaindex is greater than $t$. For these queries, this reduces resource usage for solvable queries and prevents over-allocation to unsolvable ones (red dots above the orange line), without sacrificing accuracy.

**Pareto-frontier allocation.** While thresholding offers a coarse-grained control, a more nuanced approach is a Pareto-frontier allocation. Instead of a binary stop/go decision based on a single early checkpoint, this policy aims to assign a dynamically tailored computational budget to each query based on its evolving certaindex throughout the reasoning process. The underlying principle is to leverage the observed correlation to inform a more continuous resource allocation. Rather than simply "fitting a curve", we establish an empirically-derived relationship that maps different certaindex values to an estimated optimal remaining token budget for that level of certainty. This mapping (conceptually represented by the green curve in Figure 12) reflects a trade-off: for higher certaindex values, the optimal additional budget is small, while for lower certaindex values, more computation might be warranted, up to a certain point. This allows for a dynamic allocation scheme where the maximum additional tokens for a query are continuously adjusted or capped based on its current certaindex. Queries exhibiting high certaindex (signaling proximity to a stable solution) are allocated fewer future tokens, while those still showing lower certaindex (indicating ongoing exploration or difficulty) might be allowed more, guided by this learned relationship. This strategy seeks to optimize resource use more globally across all queries, distributing the computational budget more effectively, as further elaborated in our "smart scheduler" design (§H.4).

### C.3    Comparing Certaindex with Other Signals

One may ask if there exist other signals better than certaindex. We compare certaindex with two alternative metrics for estimating resource needs. (1) *Reasoning path length*: Longer reasoning paths (more tokens) often indicate harder problems, suggesting a potential correlation between path length and compute needs (e.g., more reasoning paths). (2) *Mean normalized log probability*: Established as a measure of LLM confidence [24], higher log probabilities may correlate fewer samples needed for correct answers in SC.

Using the (SC, GSM8K, Llama3.1-8B-instruct) setup, we evaluated certaindex alongside these metrics. Four metrics were tested: certaindex's entropy measure $\mathcal{H}$, mean output length [40], mean normalized log probability [24], and their linear combinations. As shown in Fig. 13, $\mathcal{H}$ achieved the strongest correlation with ground truth compute requirements (Pearson Correlation of 0.68, Kendall's Tau of 0.61), outperforming other metrics and matching complex combinations. These results confirm that certaindex is a simple yet effective proxy for estimating inference computational demands, offering robust performance across tasks and models. Our end-to-end *token-to-accuracy* evaluations in Appendix H compare different signals for resource allocation, confirming certaindex's superior performance.

## D    Characterizing the Relationship Between Certaindex and Detection Steps

Fig. 14 demonstrates the relationship between certaindex values and program steps across different detection points in a single run, using SC on the GSM8K dataset with Llama3.1 8B Instruct model. The figure consists of six subplots, each representing a different detection step ranging from 5 to 30, indicated by "Detect @knob" values.

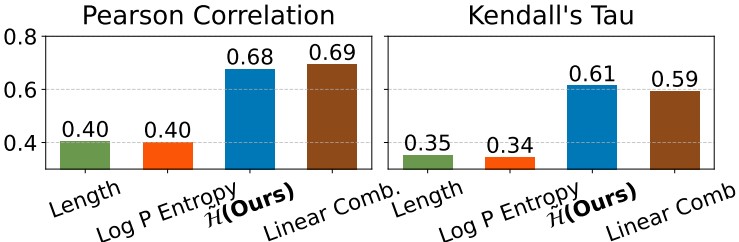

Figure 13: Correlation between certainty measurements and mean steps required to solve problems on solvable problems. We obtain the ground-truth mean steps by solving the queries using the LLM multiple times and counting the average steps.

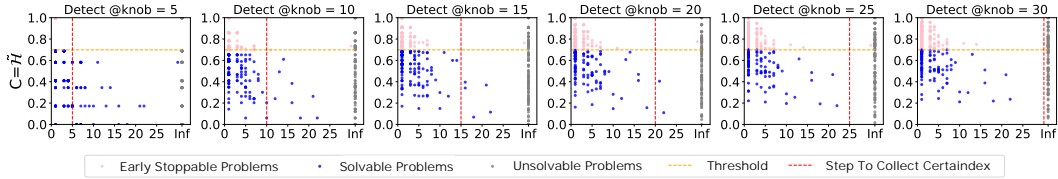

Figure 14: Certaindex Values Across Different Detection Steps in Self-Consistency Reasoning

Each point in the plots represents an individual problem, categorized into three types: Early Stoppable Problems (pink), Solvable Problems (blue), and Unsolvable Problems (gray). The x-axis shows the number of steps taken, while the y-axis displays the certaindex value ($C = \tilde{\mathcal{H}}$). A horizontal orange dashed line indicates the certaindex threshold value, and a vertical red dashed line marks the step at which certaindex is collected.

The consistent pattern across all detection points demonstrates a strong correlation between certaindex and required reasoning steps, with Pearson correlation coefficients exceeding 0.5 for solvable problems. As detection steps increase from 5 to 30, we observe a 30% increase in early-stopped problems, indicating that higher certaindex values reliably predict when the LLM is converging toward a final answer. Certaindex works as a reliable proxy for reasoning progress. However, this improved accuracy trades off against potential compute savings, as later detection points leave less opportunity for early termination. Despite this tradeoff, certaindex proves to be a reliable predictor of required reasoning steps across all detection timings, maintaining its effectiveness regardless of measurement timing.

# E   Dynasor: System Design

Dynasor is straightforward to use. Developers define reasoning programs through the provided abstraction, implementing certaindex and scaling knob control functions. These programs are submitted to the application runtime, which monitors certaindex values, dynamically adjusts resource allocation, and forwards requests to the system runtime for execution. Programs either scale up for further computation or terminate early when resources are no longer allocated.

Figure 15a illustrates Dynasor's three-component architecture: (a) a Reasoning Program Abstraction (`Program`) offering a standardized interface for diverse reasoning algorithms, (b) an Application Runtime that dynamically allocates computational resources based on CertaIndex measurements, and (c) a System Runtime managing request-level scheduling on the backend infrastructure.

This architecture enables Dynasor to adapt computation dynamically to each query's difficulty level, allocating resources proportionally to reasoning complexity rather than using static, one-size-fits-all token budgets.

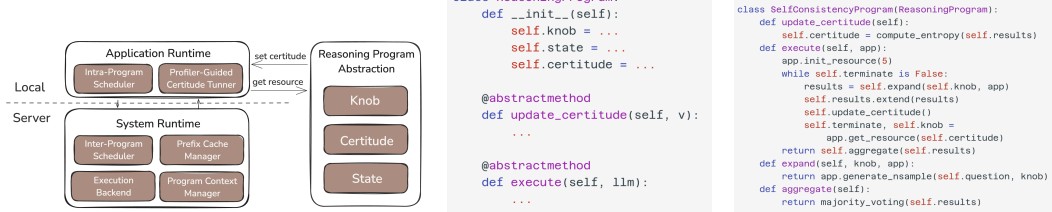

Figure 15: Left(a): Dynasor Architecture. Middle(b): Reasoning Program Interface. Right(c): Example Program (SC).

## E.1 Reasoning Program Abstraction

A reasoning program maintains three runtime properties: (1) *certaindex*, the certainty measure of the reasoning progress; (2) *knob*, the intrinsic scaling factor of the program; and (3) *state*, which stores the intermediate variables and results in previous steps.

The Reasoning Program Abstraction (or `Program`) provides a unified framework for developers to define a variety of reasoning algorithms. It introduces a narrow interface for the program to interact with the application runtime. Fig. 15a shows the structure of a reasoning program. Developers only implement (1) `update_certaindex()`, which calculates and updates the certaindex at a particular inference step, and (2) `execute()`, which runs the reasoning algorithm (Fig. 15b).

Fig. 15c illustrate a simple example implementation of SC: the `update_certaindex()` function computes the entropy across different branches, and the `execute()` function iteratively expands the generation. After each iteration, it aggregates the result and update certaindex. This process iterates until the program depletes its resources or exits.

## E.2 Application Runtime

### E.2.1 Certaindex-Based Intra-Program Scheduler

The certaindex-based intra-program scheduler controls the resource allocation of the program at runtime. The scheduler lifetime is described as follows:

**Initialization.** When a new program arrives, the scheduler initializes it with a predefined maximum resource cap and allocates resources for its first run. It also establishes a resource scheduling policy to guide future allocations.

**Resource Allocation.** The scheduler continuously monitors each program's certaindex and uses it to determine resource allocation. As programs run, they repeatedly request resources from the scheduler while updating their certaindex to reflect the progress of reasoning. When multiple programs run concurrently, the scheduler can prioritize resource allocation between them via certaindex-based intra-program allocation policy.

**Termination.** When a program's certaindex exceeds a limit based on its allocation policy or reaches the maximum resource cap, the scheduler denies further resources allocation for this program. The program then receives a termination signal and aggregates results based on its generation history.

We use the SC in Fig. 15c to illustrate the scheduler behavior. The program is first submitted to the application runtime, where the intra-program scheduler is initialized and assigned the initial resource (5 branches) to the program. SC sets a simple certaindex threshold to determine whether the program should terminate. As the program expand, it updates certaindex, and request resource from the scheduler for next iteration. The scheduler checks the certaindex against the policy, and decide if it should allocate resource for the program to continue running.

**Resource allocation policy**. A resource allocation policy determines compute allocation (e.g., branches in SC and iterations in MCTS) for each `program` based on its certaindex. §C.2 discussed two certaindex-based allocation policies: simple thresholding and pareto-frontier, both implemented in the system. Additional alternatives are analyzed in §H.4. Developers can easily configure the intra-program scheduler to use other certaindex-based policies.

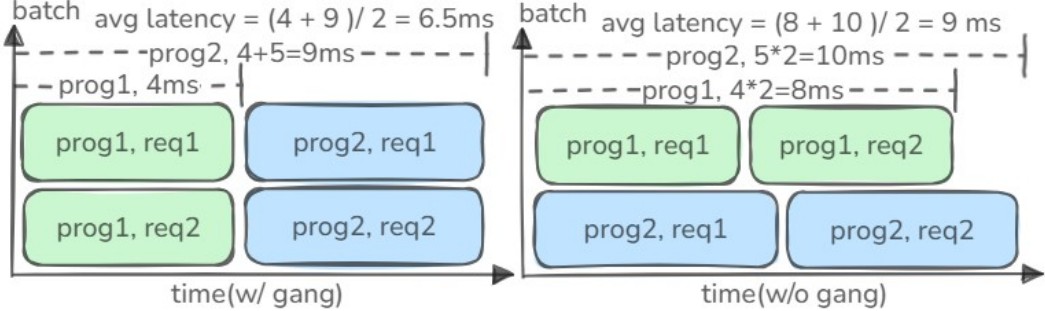

Figure 16: Illustration of Gang Scheduling

### E.2.2 Profiler-Guided Policy Calibration

We explain how to determine (or calibrate) a resource allocation policy, focusing on a certaindex threshold policy as our example. Determining an effective resource allocation policy is critical for reasoning program performance (§H.4). Program submitters often lack insight into runtime patterns, and program characteristics may shift due to algorithmic changes or data distribution shifts.

Dynasor provides an optional profile-based hyperparameter tuner to help users identify optimal scheduling policies for certaindex-based resource allocation. Users submit a batch of programs with labeled data - such as verified answers to questions (e.g., math problem solutions), response rankings, or reward model scores. The profiler collects runtime metrics, including certaindex and token usage, to determine optimal resource allocation across different certaindex ranges while maintaining accuracy requirements. For threshold-based allocation, the profiler stops allocating resources when a program's certaindex exceeds the threshold, and otherwise sets resources to a maximum cap. The threshold is calibrated to meet the accuracy requirements (e.g., not hurt accuracy in all calibration data points). This calibration process can be periodically executed in real-world serving scenarios to meet the dynamics of data distribution shifts.

## E.3 System Runtime

### E.3.1 Program-Aware Inter-Program Scheduler

The Program-Aware Inter-Program Scheduler optimizes scheduling and memory management at the program level, reducing per-program latency through two key strategies: (1) gang scheduling algorithm to prioritize requests originated from the same program, and (2) approximate Shortest-Job-First (SJF) scheduling algorithm to reduce head-of-the-line (HoL) blocking and improve per-request latency.

**Gang Scheduling.** Gang scheduling groups requests from the same program together to minimize stragglers and reduce overall completion time. Fig. 16 demonstrates the advantage of Gang scheduling over sequential scheduling using an example with two programs arriving at t=0, where the system has a batch size of 2. Each program has two requests: program 1's requests take 4 ms each, and program 2's take 5 ms each. By prioritizing one program at a time, Gang scheduling (left) reduces average latency from 9 ms to 6.5 ms compared to sequential scheduling (right).

**Approximating Shortest Job First (SJF).** Our inter-program scheduler implements an approximating SJF scheduling algorithm to mitigate HoL blocking and improve average per-program latency. A program's total execution time depends on two key factors: total compute requirements (knobs, e.g., number of branches/iterations) and token length per branch/iteration. While per branch/iteration's exact LLM generation lengths cannot be known in advance [40], we can estimate token length per iteration by leveraging program locality and using historical averages from previous iterations. This estimation, combined with the compute requirements, helps predict total execution time. In Dynasor, certaindex controls the total compute requirements (deciding how many iterations to run), while SJF uses the estimated token length per iteration to optimize execution order.

**Starvation Prevention.** Dynasor promotes *finish-time fairness* [81], which compares a program's completion time in a shared system ($T_{\text{shared}}$) to its estimated independent completion time ($T_{\text{independent}}$). This metric naturally suits LLM serving as it accounts for generation length. We define finish-time

fairness as $\phi = T_{\text{shared}}/\#$output tokens in our case, which is approximately proportional to the standard finish-time fairness since $T_{\text{independent}}$ can be estimated as $k \cdot \#$output tokens when decoding dominates the generation time, where $k$ is a constant. While Gang scheduling alone significantly improves performance without causing starvation, adding SJF optimization can potentially lead to starvation of programs with longer estimated lengths. Gang scheduling alone already provides substantial performance benefits, making it a viable standalone option. For deployments using SJF, we implement a priority escalation mechanism where programs that have been waiting too long receive elevated priority, effectively preventing starvation. As demonstrated in §H.3, Dynasor promotes fairness compared to other baseline scheduling strategies.

### E.3.2 Other System Runtime Components

**Prefix Cache Manager.** Dynasor leverages prefix cache sharing. Independent branches of a program share a unified prompt KV-cache, while dependent reasoning chains reuse their longest common prefixes. For algorithms with reward models, Dynasor reuses prefixes during reward evaluation, reducing latency. The manager also handles automatic cache eviction. When memory is constrained, the KV cache of the programs with no active requests is assigned lower priority and evicted first.

**Program Context Manager** is a thin wrapper that tracks registered programs and their runtime characteristics. It provides cache eviction hints to the prefix cache manager based on program behaviors. Unlike the static program DAGs used in SGLang/ParrotServe, this component offers more flexibility by supporting dynamic generation patterns required by algorithms like MCTS and Rebase.

**Execution Backend** manages the execution of LLM requests, optimizing model performance through techniques like CUDAGraph. This layer is designed to be adaptable to various execution engines including vLLM, TensorRT, and SGLang.

### E.4 System Implementation

We build Dynasor on top of SGLang (version 0.3.3 post1). The intra-program scheduler is implemented as a Python library on the client side, while the inter-program scheduler is integrated into the server-side scheduler.

Using Dynasor, we adapt various reasoning algorithms (Appendix B) to the `Program` interface, including their custom certaindex implementations. Our system comprises approximately 1.5k lines of Python code, covering the `Program` interface, application runtime, and system runtime. Notably, the core system runtime only comprises around $\sim 500$ lines of code change, and the changes are modular and non-invasive, making it a very clean implementation into the core serving system logic.

Adapting each reasoning program requires an additional 40 to 150 lines of code to define certaindex logic and integrate with the `Program` interface. This implementation overhead is minimal compared to the original implementations of these algorithms, which spans up to 4,000 lines of code.

## F Proof of Stopping Criteria for CoT and Probing

Given the input prompt $x$, we can stop the reasoning chain $Y_1, \ldots, Y_t, Y_{t+1}$ at time $t$ if the following holds for any $T > t$.

$$P_t(Y_{t+1}|x, Y_1, \ldots, Y_t) \overset{\text{t}}{=} P_{t+1}(Y_{t+2}|x, Y_1, \ldots, Y_{t+1}) \overset{\text{t+1}}{=} \ldots \overset{\text{T}}{=} P_T(Y_{T+1}|x, Y_1, \ldots, Y_T) = P_*(Y|x). \tag{2}$$

The above definition means that adding new steps in the reasoning chain does not change the distribution of the generated answer anymore. As a result, the reasoning chain reaches the stationary distribution, $P_*(Y|x)$, which is the desired output given the prompt $x$ in the first place.

We assume that the language model will eventually converge in the chain of thought, if the answer distribution remains the same for long enough.

Formally it is expressed as follows.

**Assumption 1.** *If equality* $P_t(Y_{t+1}|x, Y_1, \ldots, Y_t) = P_{t+i}(Y_{t+i+1}|x, Y_1, \ldots, Y_{t+i})$ *holds for any* $1 \leq i \leq k^*$,

*then $P_t(Y_{t+1}|x, Y_1, \ldots, Y_t) = P_T(Y_{T+1}|x, Y_1, \ldots, Y_T) = P_*$ holds for any $T > t$. Further assume that $k^* = \mathcal{O}(M)$, where $M$ is the number of distinct output groups.*

Based on the above assumption, we propose the test criterion in Definition 2.

**Definition 1.** *Consider $n$ samples, each corresponds to $P_t, \ldots, P_{t+n}$. Denote the mixture distribution of samples $i + 1, \ldots, i + k \in [1, \ldots, n]$ to be $\bar{P}_i^{i+k} = \frac{1}{k} \sum_{j=1}^{k} P_{i+j}$.*

Let the smallest $t$ where Eq. (2) holds be $t^*$. Then for any $i, j \geq t^*$, $\bar{P}_i^{i+k} = \bar{P}_j^{j+k} = P_*$. On the other hand, if $i < t^*$, then there exists $j > i$, so that $\bar{P}_i^{i+k} \neq \bar{P}_j^{j+k}$. This intuition is formally expressed in Lemma 2 below. We therefore propose to test the difference among $\bar{P}_i^{i+k}, \bar{P}_{i+1}^{i+k+1}, \ldots, \bar{P}_{i+k}^{i+2k}$.

**Lemma 2.** *If $\mathrm{TV}(\bar{P}_i^{i+k}, \bar{P}_{i+j}^{i+j+k}) = 0$, for any $1 \leq j \leq k$, and if $\mathrm{TV}(\bar{P}_i^{i+k-1}, \bar{P}_{i+j}^{i+j+k-1}) = 0$, for any $1 \leq j \leq k - 1$, then Eq. (2) holds for any $t \geq i$, and $T < 2k - 1$.*

*Proof.* We first have from the definition of the TV distance that

$$\left\| \bar{P}_i^{i+k} - \bar{P}_{i+j}^{i+j+k} \right\|_1 = 0.$$

For a general $1 \leq j \leq k$, we have from the definition of $\bar{P}$ that

$$\frac{1}{k} \left\| \sum_{l=1}^{j} (P_{i+l} - P_{i+l+k}) \right\|_1 = \left\| \bar{P}_i^{i+k} - \bar{P}_{i+j}^{i+j+k} \right\|_1 = 0.$$

Therefore, $\sum_{l=1}^{j} P_{i+l} = \sum_{l=1}^{j} P_{i+l+k}$.

Take $j = 1$, then

$P_{i+1} = P_{i+1+k}$. Plugging into the case where $j = 2$, then we have $P_{i+2} = P_{i+2+k}$. Expanding the recursion for all $1 \leq j \leq k$, we obtain that

$$P_{i+j} = P_{i+j+k}, \quad \forall 1 \leq j \leq k. \tag{3}$$

Similarly from the assumption that $\mathrm{TV}\left(\bar{P}_i^{i+k-1}, \bar{P}_{i+j}^{i+j+k-1}\right) = 0$, for any $1 \leq j \leq k - 1$, we have that:

$$0 = \left\| \bar{P}_i^{i+k-1} - \bar{P}_{i+j}^{i+j+k-1} \right\|_1 = \frac{1}{k-1} \left\| \sum_{l=1}^{j} (P_{i+l} - P_{i+l+k-1}) \right\|_1, \quad \forall 1 \leq j \leq k - 1,$$

and consequently that

$$P_{i+j} = P_{i+j+k-1}, \quad \forall 1 \leq j \leq k - 1. \tag{4}$$

The equalities in Eqs. (3) and (4) form a graph over the vertices that are $P_{i+1}, \ldots, P_{i+2k-1}$, where the equalities are the edges. Note that all the nodes are edge connected by virtue of Eqs. (3) and (4). Therefore, we have that

$$P_j = P_l, \quad \forall i + 1 \leq j, l \leq 2k - 1.$$

$\square$

We therefore define the approximate stopping criteria as follows.

**Definition 2.** *We define the $\epsilon$-accuracy of the stopping criteria:*

$$\mathrm{TV}(\bar{P}_i^{i+k}, \bar{P}_{i+j}^{i+j+k}) \leq \epsilon, \text{ for any } 1 \leq j \leq k;$$
$$\mathrm{TV}(\bar{P}_i^{i+k-1}, \bar{P}_{i+j}^{i+j+k-1}) \leq \epsilon, \text{ for any } 1 \leq j \leq k - 1.$$

We then ask: how many data samples $k$ do we need, to be able to test our stopping criteria up to $\epsilon$-accuracy? More concretely, we construct the estimator of $\bar{P}_l^{l+t}$, for $l = i + 1, \ldots, i + k$, and $t = k - 1, k$ as follows. Assume we have $M$ disjoint groups of the outputs. Denote each group of answer as $C_m$, for $m = 1, \ldots, M$. Then we estimate the average probability mass

function $\bar{P}_l^{l+t}$ by defining the Bernoulli random variable for each $C_m$: $Z_\tau^{C_m} = \mathbb{1}\{Y_\tau \in C_m\}$, and $\hat{P}_l^{l+t}(S) = \frac{1}{t}\sum_{\tau=l+1}^{l+t} Z_\tau^S$.

The next lemma establishes that with $k = \widetilde{\Omega}\left(\frac{M+\log(1/\delta)}{\epsilon^2}\right)$, then $\mathrm{TV}(\bar{P}_l^{l+t}, \hat{P}_l^{l+t}) \le \epsilon/3$, for all $l = i+1, \ldots, i+k$, and $t = k-1, k$, with $1-\delta$ probability. Then we just have to test that

$$\mathrm{TV}(\hat{P}_i^{i+k}, \hat{P}_{i+j}^{i+j+k}) \le \epsilon/3, \text{ for any } 1 \le j \le k;$$
$$\mathrm{TV}(\hat{P}_i^{i+k-1}, \hat{P}_{i+j}^{i+j+k-1}) \le \epsilon/3, \text{ for any } 1 \le j \le k-1.$$

If the above inequalities stand with the established number of steps $k$, then by the triangle inequality of the total variation (TV) distance, we obtain the $\epsilon$-accuracy in the stopping criteria in Definition 2:

$$\mathrm{TV}(\bar{P}_i^{i+k}, \bar{P}_{i+j}^{i+j+k}) \le \mathrm{TV}(\bar{P}_i^{i+k}, \hat{P}_i^{i+k}) + \mathrm{TV}(\hat{P}_i^{i+k}, \hat{P}_{i+j}^{i+j+k}) + \mathrm{TV}(\hat{P}_{i+j}^{i+j+k}, \bar{P}_{i+j}^{i+j+k}) \le \epsilon.$$

**Lemma 1.** *If* $k = \widetilde{\Omega}\left(\frac{M+\log(1/\delta)}{\epsilon^2}\right)$, *then* $\mathrm{TV}(\bar{P}_l^{l+t}, \hat{P}_l^{l+t}) \le \epsilon/3$, *for all* $l = i+1, \ldots, i+k$, *and* $t = k-1, k$, *with* $1-\delta$ *probability.*

*Proof.* To prove convergence in the total variation (TV) distance, we first express it as follows. Let $S$ belong to the power set (denoted as $\exp^C$) of $\bigcup_{m=1}^{M} C_m$. Then

$$\mathrm{TV}(\bar{P}, \hat{P}) = \sup_{S\in\exp^C} \left| \bar{P}(S) - \hat{P}(S) \right|.$$

We then prove convergence for a fixed set $S$, and then apply union bound to $\exp^C$ to obtain the final result.

Denote a Bernoulli random variable $Z_\tau^S = \mathbb{1}\{Y_\tau \in S\}$. Then $\hat{P}_l^{l+t}(S) = \frac{1}{t}\sum_{\tau=l+1}^{l+t} Z_\tau^S$. On the other hand, $\bar{P}_l^{l+t}(S) = \frac{1}{t}\sum_{\tau=l+1}^{l+t} P_\tau(Y_\tau \in S | x, Y_1, \ldots, Y_{\tau-1})$,

Note that $Y_\tau | x, Y_1, \ldots, Y_{\tau-1} \sim P_\tau(\cdot | x, Y_1, \ldots, Y_{\tau-1})$. Therefore,

$$\mathbb{E}\left[Z_\tau^S | x, Y_1, \ldots, Y_{\tau-1}\right] = P_\tau(Y_\tau \in S | x, Y_1, \ldots, Y_{\tau-1}).$$

This means that $Z_\tau^S - P_\tau(Y_\tau \in S | x, Y_1, \ldots, Y_{\tau-1})$ is a martingale difference sequence. We also know that $\left|Z_\tau^S - P_\tau(Y_\tau \in S | x, Y_1, \ldots, Y_{\tau-1})\right| \le 1$. Therefore, by the Azuma-Hoeffding inequality, we have:

$$\mathbb{P}\left(\left|\sum_{\tau=l+1}^{l+t}\left(Z_\tau^S - P_\tau(Y_\tau \in S | x, Y_1, \ldots, Y_{\tau-1})\right)\right| \ge \epsilon\right) \le 2\exp\left(-\frac{\epsilon^2}{2t}\right),$$

or

$$\mathbb{P}\left(\left|\bar{P}_l^{l+t}(S) - \hat{P}_l^{l+t}(S)\right| \ge \epsilon\right) = \mathbb{P}\left(\left|\frac{1}{t}\sum_{\tau=l+1}^{l+t} Z_\tau^S - \frac{1}{t}\sum_{\tau=l+1}^{l+t} P_\tau(Y_\tau \in S | x, Y_1, \ldots, Y_{\tau-1})\right| \ge \epsilon\right)$$
$$\le 2\exp\left(-\frac{t\epsilon^2}{2}\right).$$

Taking a union bound over $S \in \exp^C$, we have:

$$\mathbb{P}\left(\mathrm{TV}(\bar{P}_l^{l+t}, \hat{P}_l^{l+t}) \ge \epsilon\right) = \mathbb{P}\left(\sup_{S\in\exp^C}\left|\bar{P}_l^{l+t}(S) - \hat{P}_l^{l+t}(S)\right| \ge \epsilon\right)$$
$$\le 2^M \mathbb{P}\left(\left|\bar{P}_l^{l+t}(S) - \hat{P}_l^{l+t}(S)\right| \ge \epsilon\right) \le 2^{M+1}\exp\left(-\frac{t\epsilon^2}{2}\right).$$

On top of that, taking a union bound over $l = i+1, \ldots, i+k$, and $t = k-1, k$, we have that the probability of $\mathrm{TV}(\bar{P}_l^{l+t}, \hat{P}_l^{l+t}) \ge \epsilon$ for one of the combination of $l = i+1, \ldots, i+k$, and $t = k-1, k$ is less than:

$$\sum_{l=i+1}^{k}\sum_{t=k-1}^{k} \mathbb{P}\left(\mathrm{TV}(\bar{P}_l^{l+t}, \hat{P}_l^{l+t}) \ge \epsilon\right) \le 2^{M+2}k \cdot \exp\left(-\frac{(k-1)\epsilon^2}{2}\right).$$

Table 1: Offline workload Configurations. LiveCodeBench (LCB), Llama2 7B [5], Skywork7B [82], Llemma 7B and 34B [13] are fine-tuned models used in different settings as LLM/reward model.

| (Algo., Dataset, LLM) | Reward Model | # Samples | Resource Cap |
|---|---|---|---|
| (SC, LCB, Llama3.1 8B) | / | 400 | 5,10,15,20,25,30 |
| (SC, GSM8K, Llama3.1 8B) | / | 1000 | 5,10,15,20,25,30 |
| (MCTS, ASDiv, Llama2 7B) | Skywork 7B | 300 | 3,7,10,15,20 |
| (MCTS, GSM8K, Llama2 7B) | Skywork 7B | 300 | 3,7,10,15,20 |
| (Rebase, MATH, Llemma 7B) | Llemma 34b | 500 | 16,32,64,128 |
| (Rebase, GSM8K, Llemma 34B) | Llemma 34b | 500 | 16,32,64,128 |

Table 2: Online workload Configurations

| Algorithm | Dataset | LLM | Reward Model | Base Deadline (s) |
|---|---|---|---|---|
| SC | MATH | Llama3.1 8B | / | 240 |
| MCTS | ASDiv | Llama2 7B | Skywork 7B | 60 |
| Rebase | GSM8K | Llemma 34B | Llemma 34b | 300 |

In other words, whenever

$$k = \widetilde{\Omega}\left(\frac{M + \log(1/\delta)}{\epsilon^2}\right),$$

we have $\text{TV}(\bar{P}_l^{l+t}, \hat{P}_l^{l+t}) \leq \epsilon/3$, for all $l = i+1, \ldots, i+k$, and $t = k-1, k$, with $1 - \delta$ probability. Here $\widetilde{\Omega}(\cdot)$ means that we omit logarithmic order terms in $M$ and $\epsilon$.

$\square$

# G   Detailed Setups for Offline and Online Experiments

Table 3: Hyperparameter configurations for certaindex. Rebase is evaluated on the MATH-OAI [38] subset of the MATH benchmark.

| Algorithm | Dataset | Thres. ($\tilde{\mathcal{H}}_\tau$) | Thres. ($\mathcal{R}_\tau$) | Detect @knob |
|---|---|---|---|---|
| SC | MATH | 0.7 | / | 5 |
| SC | GSM8K | 0.7 | / | 5 |
| SC | LiveCodeBench | 0.4 | / | 5 |
| MCTS | GSM8K | 0.99 | 0.4 | 3 |
| MCTS | ASDiv | / | 0.4 | 3 |
| Rebase | GSM8K | 0.85 | 0.99 | 16 |
| Rebase | MATH | 0.75 | / | 16 |

**Compute.** All experiments run on a GPU cluster (Runpod) equipped with A100 (80GB) GPUs. GPU usage varies by method: Rebase and MCTS use two GPUs to separately serve their LLM and reward models, while SC requires only one GPU for request processing. For final answer selection, SC uses simple majority voting, Rebase applies weighted majority voting, and MCTS selects the best path using a reward model via best-of-n.

**General settings.** Tables 1 and 2 summarize our offline and online evaluation workloads, respectively. Table 3 lists the certaindex hyperparameters for each workload, including the thresholds at the detection step (`detect @knob`). A program terminates at the detection step if its certaindex values meet all threshold conditions.

## G.1   Batch Inference Setup: CoT

**Metrics.** For offline workloads, we measure *tokens-to-accuracy*, the total number of generated tokens needed to achieve specific accuracy levels across reasoning queries. This metric reflects the accuracy-cost trade-off, critical for end users as LLM platforms charge based on token usage. Table 1 details the offline experimental settings for SC/MCTS/Rebase. Each problem is allocated a maximum computation budget (parameterized by a Resource Cap) defining the sampling limit for SC and iteration limit for MCTS. Resource caps correspond to different points in Fig. 7. While

scheduling methods may terminate programs early, they cannot exceed these caps. Query deadlines are calculated as the product of three factors: the SLO scale, the query's difficulty factor, and a base deadline, reflecting the additional time required for more challenging problems.

We evaluate the Certaindex-based early termination method (§2.1) against baseline uniform token allocation across multiple scales of distilled DeepSeek models (7B, 14B, and 32B) [14] on mathematical reasoning benchmarks AIME24 and AMC23 [20], and MATH500 [38]. Unlike the baselines that uniformly increases token budgets, our appraoch early terminates by monitoring Certaindex at various intervals ($T = 32$, 64, 128. 256, and 320). All requests are given max token budget of 16K. For each interval, we vary the early termination parameter $N$ (the required number of consecutive consistent answers), generating different points along each line. For fair comparison, appropriate accuracy thresholds were calibrated to model scale - with 32B models evaluated against stricter thresholds above QwQ [33] levels and reduced thresholds for smaller models - while setting higher targets for simpler tasks where greater accuracy is achievable.

### G.2   Batch Inference Setup: SC, MCTS, Rebase

**Metrics.** For online workloads, we simulate query arrivals using dataset queries and assign deadlines to each query. System performance is evaluated by *P90 deadline attainment*, the percentage of queries completed within their deadlines. We vary *arrival rates* and *SLO scales* to assess system performance under different conditions. Tab. 2 outlines the online experiment settings. Request arrivals follow a Poisson process with varying rates. Deadlines are difficulty-aware, determined using a simple policy: oracle difficulty for each query is estimated through extensive trial runs ($> 100$) for each algorithm-dataset combination, classifying queries as always correct (difficulty factor $= 1$), always incorrect ($= 3$), or variable ($= 2$). Deadlines are calculated as the product of the SLO scale, the difficulty factor, and a base deadline, accounting for the additional time required for more challenging problems.

**Baselines**. For offline evaluation, we compare Dynasor against a modified SGLang intra-program scheduler using the following policies:

• (1) **baseline-even**, allocates resources uniformly across all reasoning programs using the resource cap settings in Tab.1. Different cap values result in different accuracy levels, as larger caps allow more computation for all problems. Specifically, for each algorithm, all queries receive identical resources: SC uses the same number of branches, Rebase uses the same width, and MCTS uses the same number of search iterations, as specified in Tab.1. This baseline reflects the behavior of reasoning programs in current systems: user may allocate equal resources on all problems.

• (2) **baseline-length.** This policy uses the cumulative token count generated at a specific step (specified as Detect@knob in Table 3) as the program's progress signal. The detection step matches the one used in the certaindex-based approach. The scheduler either continues allocating resources or terminates the program based on a predefined token-length threshold.

Output token length is commonly used as a scheduling indicator in existing LLM serving systems [40; 76; 83]. It also correlates strongly with the compute requirements (knob size) as we have discussed in §C.3.

### G.3   Online Inference Detailed Setup

**Baselines.** For online evaluation, we evaluate Dynasor against modified SGLang with different inter-program schedulers on P90 deadline attainment.

• **SGLang** [34]. SGLang represents a strong baseline as it incorporates key optimizations in LLM serving: its longest-prefix matching (LPM) algorithm efficiently batches requests with common prefixes, while prefix caching reduces redundant computation. We enabled these optimizations and tuned for stable performance with 70% memory utilization.

• **Parrot** [39]. Parrot uses gang scheduling (App-FIFO) to prioritize requests in the same program. We implemented its scheduler on top of SGLang for fair comaprison. By grouping requests from the same program together, it minimizes context switches and maximizes KV cache reuse across batches.

For experiments (a) and (b), we use fixed resource cap settings (SC: 20, MCTS: 15, Rebase: 128) to compare Dynasor with baselines, which lack adaptive compute scheduling for LLM reasoning

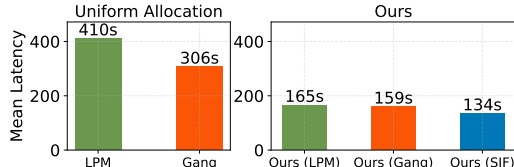

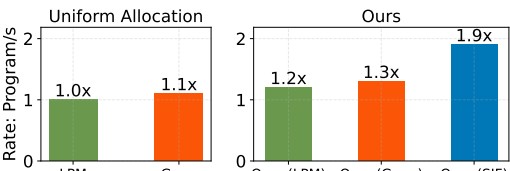

Figure 18: Performance improvement break-down in online SC (GSM8K): Impact of gang scheduling, certaindex-based allocation, and SJF on mean latency with fixed request rate (rps = 16).

Figure 19: Performance improvement break-down of online self-consistency (MATH) under fixed P90 SLO constraints.

programs. We measure SLO attainment while *ensuring accuracy is maintained*. Our method employs the same resource allocation policy validated in §4.1 and §I, which preserves accuracy. In experiment (c), we explore the relationship between achievable accuracy and SLO attainment by varying resource cap settings.

# H  Other Ablation Studies

## H.1  Effectiveness of CoT in Deepseek-R1

To validate scalability, we extended our experiments to the larger DeepSeek-R1 model on AIME and AMC datasets (Figure 17). The results align with our findings from smaller distill models, demonstrating consistent efficiency gains: DeepSeek-R1 achieves 12% token savings on AIME problems and 24% on AMC problems while maintaining baseline accuracy levels.

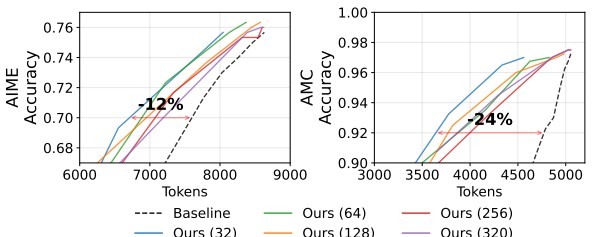

Figure 17: Apply Dynasor on DeepSeek-R1

## H.2  Ablation: Scheduling Component Contributions

Figures 18 and 19 show the contribution of different components using SC on GSM8K and MATH datasets, measuring mean latency and maximum sustainable rate under the same P90 attainment. Using LPM (SGLang) as the baseline, we analyze three factors: gang scheduling, SJF, and certaindex-based resource management. Gang scheduling improves latency and throughput in both uniform and certaindex-aware resource allocation by ensuring the requests in the same program are scheduled together, reducing resource fragmentation and context switching overhead. For GSM8K workload, certaindex-based resource management dominates the improvements, reducing mean latency by up to 60% through token savings of up to 50%, while Gang scheduling and SJF contributions are more modest. For MATH workload, certaindex-aware allocation achieves a 1.2x peak rate through early termination of low-value computations (as only 10% tokens are saved on MATH), while SJF provides the most significant gain with a 1.9x peak rate by prioritizing shorter jobs and reducing head-of-line blocking. These results validate our two-level scheduling approach: intra-program optimization through gang scheduling and certaindex-aware allocation, combined with inter-program optimization through SJF.

## H.3  Fairness Analysis

We apply finish-time fairness (§E.3.1) as the fairness metric and use latency / number of tokens as a proxy. Figure 20 shows the comparison on certaindex-based resource scheduling, gang scheduling, and SJF on finish-time fairness on MATH using SC with a rate of 8 pps using Llama3.1 8B. Gang scheduling shows consistent fairness improvement compared to non-gang scheduling. This shows that prioritizing existing programs can improve finish-time fairness. Certaindex-based resource allocation also consistently improve finish-time fairness due to resource cutting by the intra-program scheduler. Adding SJF shows fairness improvement at the later 50% fraction of the job compared to without

Table 4: Token consumption comparison of different scheduling strategies while maintaining accuracy

| Allocation Method | SC/MATH | MCTS/ASDiv |
|---|---|---|
| Baseline | 1.180M | 354K |
| **Static Thres. (Ours)** | 1.05M (-11.0%) | 308K (-13.0%) |
| **+ Initial Step Curve Fit.(Ours)** | 1.04M (-11.9%) | 306K (-13.6%) |
| **+ 5-Step Thres. (Ours)** | 1.03M (-12.7%) | 307K (-13.3%) |
| **+ Single-Step Thres. (Ours)** | 1.03M (-12.7%) | 306K (-13.6%) |
| **+ Dynamic Curve Fitting (Ours)** | 1.01M (-14.4%) | 298K (-15.8%) |

SJF, and at the later 35% fraction of job compared to LPM. In all case, SJF shows the fairness metric no worse than even resource allocation.

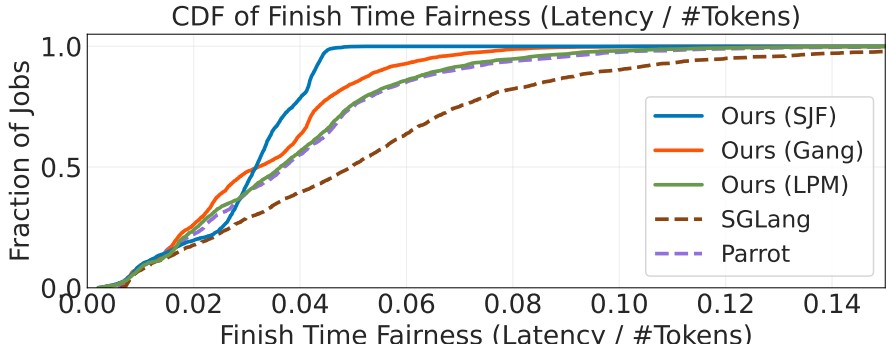

Figure 20: Finish-Time Fairness.

## H.4    Fine-grained Resource Allocation

We compare simple thresholding against fine-grained resource allocation in this section. §4.1 and §4.2 adopt a simple *static threshold* mechanism: extract certaindex at a fixed step and decide if to terminate program based of the value of certaindex. While effective, there is room for optimization through more sophisticated scheduling strategies. To explore this, we conducted experiments in offline batch processing using the setting (SC, MATH) and (MCTS, ASDiv), both with a maximum step limit of 20 as per-query resource cap, to evaluate more complex allocation strategies that enable early termination while maintaining accuracy. The results, measured in token savings, are presented in Tab. 4. For the static threshold, we collected certaindex and apply the threshold in Tab. 3.

We first examine an *Initial Step Curve Fitting* approach, which fits a skyline curve (illustrated by the green lines in Fig. 12) using certaindex values collected at the same steps as the static threshold. Although this method enables finer-grained resource allocation for varying certaindex values, it relied on early certaindex values, which maybe inaccruate as reasoning progresses, resulting in marginal improvements (less than 1% compute savings) compared to simple static thresholding.

To enhance prediction accuracy, we test more frequent certaindex collection: every 5 steps (*5-Step Thres.*) and every step (*Single-Step Thres.*), based on which we similarly apply threshold filtering or skyline curve fitting at every step (*Dynamic Curve Fitting*). These approaches achieve higher token savings (up to 3.4% over static threshold). However, implementing them introduces scheduling trade-offs in real-world deployment due to their impact on scheduling and parallelism. Specifically, frequent certaindex collection may disrupt the concurrent execution of reasoning programs. For instance, in SC, instead of running 20 samples concurrently, we must process them in sequential batches (e.g., 4 batches of 5 samples) to track the certaindex. This shift from parallel to sequential execution substantially increases latency. Our benchmark shows an increase from 289s to 366s in mean latency when serving 500 programs. Given these practical constraints, we opt to implement the simple static threshold in our end-to-end experiments, prioritizing system performance over marginal token savings; but note in applications which prioritize cost over latency, such allocation strategies remain effective and are implemented in Dynasor.

# I  *Token-to-accuracy* Performance of Dynasor on MATH using SC

Figure 21 presents the *token-to-accuracy* results using a Llama3.1 8B Instruct model, where resources are allocated by Dynasor using the threshold and detect @knob configurations described in Tab. 3. Our proposed method achieves the same accuracy while reducing computational costs by 11%.

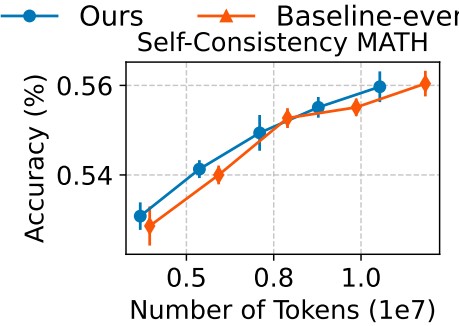

Figure 21: Token-to-accuracy Performance Using SC on MATH

# J  Probe Prompt Sensitivity

We conducted comprehensive ablation studies using three different probe prompts to evaluate robustness of early-exit performance to prompt phrasing in CoT.

- **P1:** `...  Oh, I suddenly got the answer to the whole problem, Final Answer\n\n[ \boxed{`
- **P2:** `Based on the reasoning so far, the answer is:  \boxed{`
- **P3:** `Final Answer\n\n[ \boxed{`

Table 5 and 6 demonstrate that different prompt styles achieve similar token savings at comparable accuracy levels. For instance, on AMC-32B at 96% accuracy, all three prompts save 20-25% tokens (P1: 22.5%, P2: 19.8%, P3: 24.9%). Similarly, on AIME-32B at 71% accuracy, the token savings range from 11-14% across all three prompts (P1: 13.9%, P2: 14.4%, P3: 11.4%). This shows the method's effectiveness is robust to prompt phrasing, with consistent performance regardless of the specific guidance style used.

Table 5: Lower Target Accuracy Scenarios. Format: Accuracy / Token Saving (%).

| Model + Dataset | Early Exit (P1) | Early Exit (P2) | Early Exit (P3) |
|---|---|---|---|
| AIME 8B | 49% / 30.4% | 49% / 32.2% | 49% / 34.9% |
| AMC 8B | 88% / 28.0% | 88% / 25.1% | 80% / 33.6% |
| AIME 32B | 69% / 22.4% | 69% / 17.8% | 69% / 16.9% |
| AMC 32B | 89% / 36.6% | 90% / 36.9% | 92% / 32.7% |

Table 6: Higher Target Accuracy Scenarios. Format: Accuracy / Token Saving (%).

| Model + Dataset | Early Exit (P1) | Early Exit (P2) | Early Exit (P3) |
|---|---|---|---|
| AIME 8B | 52% / 15.6% | 52% / 4.0% | 52% / 27.7% |
| AMC 8B | 91% / 19.4% | 90% / 14.5% | 83% / 30.9% |
| AIME 32B | 71% / 13.9% | 71% / 14.4% | 71% / 11.4% |
| AMC 32B | 96% / 22.5% | 96% / 19.8% | 96% / 24.9% |

## K    Information Completeness and Truncation Fidelity

To examine whether truncated CoT retains sufficient reasoning information, we varied the confidence threshold that governs early exits. We report the fraction of *Right→Wrong* cases, where standard decoding is correct but early exit is not.

Table 7: Trade-off between accuracy, token saving, and information loss (Right→Wrong).

| Model | Accuracy (%) | Token Saving (%) | Right→Wrong (%) |
|---|---|---|---|
| AMC 32B | 89.0 | 36.6 | 8.2 |
| | 96.2 | 22.5 | 1.0 |
| | 97.3 | 16.1 | 0.0 |
| AIME 32B | 62.0 | 39.8 | 11.7 |
| | 69.3 | 22.4 | 4.7 |
| | 72.0 | 9.7 | 1.7 |

Table 7 demonstrates that aggressive early exits show substantial Right→Wrong rates of 8.2% and 11.7% for AMC and AIME respectively, indicating significant information loss. However, as confidence thresholds become stricter, this drops dramatically from 8.2% to 1.0% to 0.0% for AMC, and from 11.7% to 4.7% to 2.7% for AIME. With sufficiently strict thresholds, the Right→Wrong rate becomes negligible, indicating that when the confidence threshold is high enough, the truncated reasoning contains nearly all necessary information for correct answers. This validates that our adjustable confidence mechanism effectively controls the trade-off between efficiency and reasoning completeness.

## L    Incorrect Stabilization

Sometimes, our method encounters incorrect stabilization in CoT, where the model converges to a wrong answer with high confidence. However, our analysis shows that with sufficiently strict confidence thresholds, our approach largely preserves the model's original reasoning capability while primarily serving as a token-saving mechanism.

The key insight is that when confidence thresholds are strict enough, our method essentially maintains the model's inherent accuracy: if the model would originally get an answer right, our method keeps it right; if the model would originally get it wrong, our method doesn't fix that underlying limitation in most cases.

Table 8 show the result. With conservative thresholds, the Right→Wrong rate drops to nearly zero (0.0% for AMC, 1.7% for AIME), while Wrong→Right remains minimal but consistent ( 0.7-1.0%). This demonstrates that our method primarily acts as an efficiency optimization rather than an accuracy enhancer. The small Wrong→Right improvements likely represent cases where early termination prevented overthinking, but this effect is marginal. Essentially, when thresholds are strict enough, we preserve the model's original problem-solving trajectory while achieving substantial token savings (9.7-16.1% even with conservative settings).

Table 8: Stabilization behavior under different thresholds.

| Model | Accuracy (%) | Token Saving (%) | Wrong→Right (%) | Right→Wrong (%) |
|---|---|---|---|---|
| AMC 32B | 89.0 | 36.6 | 0.8 | 8.2 |
| | 96.2 | 22.5 | 0.8 | 1.0 |
| | 97.3 | 16.1 | 0.8 | 0.0 |
| AIME 32B | 62.0 | 39.8 | 0.7 | 11.7 |
| | 69.3 | 22.4 | 1.0 | 4.7 |
| | 72.0 | 9.7 | 0.7 | 1.7 |

