# OpenReview forum: "Efficiently Scaling LLM Reasoning Programs with Certaindex"
_NeurIPS.cc/2025/Conference — NeurIPS 2025 poster_

### Official Review · Reviewer_nxvt · 2025-06-27

**Clarity:** 3
**Significance:** 3
**Originality:** 3
**Rating:** 5
**Confidence:** 4

**Summary:**

This paper proposes Certaindex, a lightweight, algorithm-agnostic metric designed to measure the stability of intermediate answers in LLM reasoning tasks. The authors observe that many test-time reasoning methods (e.g., CoT, MCTS) generate excessive tokens even after reaching a stable answer. To address this inefficiency, they introduce a probing technique ("Probe-in-the-Middle") that periodically prompts the model to give an intermediate answer. If these answers stabilize, the model likely will not change its final answer, allowing for early termination of generation. They generalize this insight into Certaindex, which can be used across multiple reasoning algorithms and integrated into a scheduling system (Dynasor) to dynamically allocate compute.

**Questions:**

1. Probe Prompt Design: Have you tried alternative, less dramatic probe prompts, such as "Based on the reasoning so far, the answer is: boxed"? If so, how sensitive is performance to this choice?

2. Information Completeness: Do you have quantitative or intuitive analysis on whether the truncated CoT contains all the information necessary to output the correct final answer? Are there counterexamples where early answers are stable but ultimately incorrect due to missing later reasoning steps?

3. Incorrect Stabilization: Could you provide more insight into how often the model stabilizes to an incorrect answer, and whether Certaindex can differentiate between "correct and stable" vs. "wrong and stable" situations?

**Ethical Concerns:**

["NO or VERY MINOR ethics concerns only"]

**Limitations:**

Yes.

**Quality:**

3

**Strengths And Weaknesses:**

Strengths:

- The paper identifies and addresses a compelling inefficiency in modern LLM reasoning—token overuse even when the answer is already internally determined, especially in CoT and similar paradigms.

- Across multiple reasoning methods and datasets, the proposed method achieves substantial token savings and throughput improvements without hurting accuracy.

- The implementation of Dynasor within an LLM serving system (SGLang) with minimal code change shows the practical viability of the approach.

Weaknesses:

- The probe prompt ("Oh, I suddenly got the answer to the whole problem, Final Answer: boxed") seems overly dramatic and could potentially bias the model's behavior. The paper claims the exact phrasing is not critical, but does not provide quantitative or ablation studies on alternative, less stylized prompts.

- Certainty metric is primarily based on answer repetition: While effective, it may not capture deeper uncertainty or error modes beyond output consistency, especially when models are confidently wrong. The authors attempt to filter out hesitations (e.g., “wait”), but this is still heuristic.

- Limited analysis on incorrect stabilization: While the paper emphasizes when to stop generating because the answer is stable, it does not deeply discuss when this leads to confidently wrong outputs and how often that happens, which could impact the overall utility of early stopping.

- Truncation fidelity: The underlying assumption is that truncated CoTs still contain all necessary information to reach the correct answer. While there's theoretical justification and empirical support, this aspect could benefit from more intuitive or diagnostic discussion.

---

> ### Author Rebuttal · Authors · 2025-07-29
>
> Respond to nxvt
>
> Thank you for your thoughtful review and the encouraging feedback on Certaindex and Dynasor. Below, we address your questions in detail.
>
> # Questions
>
> ## Q1 Probe Prompt Design
>
> Thank you for this important question about probe prompt sensitivity. We conducted comprehensive ablation studies using three different probe prompts to evaluate their impact on performance and token efficiency across 10 problem traces:
> - P1: "... Oh, I suddenly got the answer to the whole problem, Final Answer\n\n\[ \boxed{"
> - P2: "Based on the reasoning so far, the answer is: \boxed{"
> - P3: "Final Answer\n\n\[ \boxed{"
>
> ### Results Summary
>
> **Lower Target Accuracy Scenarios**
>
> | Model + Dataset | Early Exit (P1)    | Early Exit (P2)    | Early Exit (P3)    |
> |-----------------|--------------------|--------------------|--------------------|
> | AIME 8B         | 0.49 / 30.4%       | 0.49 / 32.2%       | 0.49 / 34.9%       |
> | AMC 8B          | 0.88 / 28.0%       | 0.88 / 25.1%       | 0.80 / 33.6%       |
> | AIME 32B        | 0.69 / 22.4%       | 0.69 / 17.8%       | 0.69 / 16.9%       |
> | AMC 32B         | 0.89 / 36.6%       | 0.90 / 36.9%       | 0.92 / 32.7%       |
>
> **Higher Target Accuracy Scenarios**
>
> | Model + Dataset | Early Exit (P1)    | Early Exit (P2)    | Early Exit (P3)    |
> |-----------------|--------------------|--------------------|--------------------|
> | AIME 8B         | 0.52 / 15.6%       | 0.52 / 4.0%        | 0.52 / 27.7%       |
> | AMC 8B          | 0.91 / 19.4%       | 0.90 / 14.5%       | 0.83 / 30.9%       |
> | AIME 32B        | 0.71 / 13.9%       | 0.71 / 14.4%       | 0.71 / 11.4%       |
> | AMC 32B         | 0.96 / 22.5%       | 0.96 / 19.8%       | 0.96 / 24.9%       |
>
> *Format: Accuracy / Token Saving %*
>
> The results demonstrate that different prompt styles achieve similar token savings at comparable accuracy levels. For instance, on AMC-32B at 96% accuracy, all three prompts save 20-25% tokens (P1: 22.5%, P2: 19.8%, P3: 24.9%). Similarly, on AIME-32B at 71% accuracy, the token savings range from 11-14% across all three prompts (P1: 13.9%, P2: 14.4%, P3: 11.4%). This shows the method's effectiveness is robust to prompt phrasing, with consistent performance regardless of the specific guidance style used.
>
>
>
> ## Q2 Information Completeness and Truncation Fidelity (weakness 4 & question 2)
>
> Yes, our method does encounter cases where early answers are stable but ultimately incorrect due to missing later reasoning steps. This is precisely why we implement adjustable confidence thresholds.
>
> Our quantitative analysis reveals a clear relationship between confidence thresholds and information completeness. The analysis shows that when confidence thresholds are more aggressive, there's significant information loss, but as thresholds become stricter, the Right→Wrong rate (cases where standard method was correct but early exit was wrong) decreases dramatically.
>
> **AMC 32B Results (Standard Baseline: 96.5% accuracy):**
>
> | Accuracy | Token Saving % | Right→Wrong % |
> |----------|----------------|---------------|
> | 0.8900   | 36.6%          | 8.2%          |
> | 0.9625   | 22.5%          | 1.0%          |
> | 0.9725   | 16.1%          | 0.0%          |
>
> **AIME 32B Results (Standard Baseline: 73.0% accuracy):**
>
> | Accuracy | Token Saving % | Right→Wrong % |
> |----------|----------------|---------------|
> | 0.6200   | 39.8%          | 11.7%         |
> | 0.6933   | 22.4%          | 4.7%          |
> | 0.7200   | 9.7%          | 1.7%          |
>
>
> The data demonstrates that aggressive early exits show substantial Right→Wrong rates of 8.2% and 11.7% for AMC and AIME respectively, indicating significant information loss. However, as confidence thresholds become stricter, this drops dramatically - from 8.2% to 1.0% to 0.0% for AMC, and from 11.7% to 4.7% to 2.7% for AIME. With sufficiently strict thresholds, the Right→Wrong rate becomes negligible, indicating that when the confidence threshold is high enough, the truncated reasoning contains nearly all necessary information for correct answers. This validates that our adjustable confidence mechanism effectively controls the trade-off between efficiency and reasoning completeness.
>
> ## Q3 Incorrect Stabilization
>
> Our method does encounter incorrect stabilization, where the model converges to a wrong answer with high confidence. However, our analysis shows that with sufficiently strict confidence thresholds, our approach largely preserves the model's original reasoning capability while primarily serving as a token-saving mechanism.
>
> The key insight is that when confidence thresholds are strict enough, our method essentially maintains the model's inherent accuracy - if the model would originally get an answer right, our method keeps it right; if the model would originally get it wrong, our method doesn't fix that underlying limitation.
>
> **AMC 32B Results (Standard Baseline: 96.5% accuracy):**
>
> | Early Exit Accuracy | Token Saving % | Wrong→Right % | Right→Wrong % |
> |---------------------|----------------|---------------|---------------|
> | 0.8900              | 36.6%          | 0.8%          | 8.2%          |
> | 0.9625              | 22.5%          | 0.8%          | 1.0%          |
> | 0.9725              | 16.1%          | 0.8%          | 0.0%          |
>
>
> **AIME 32B Results (Standard Baseline: 73.0% accuracy):**
>
> | Early Exit Accuracy | Token Saving % | Wrong→Right % | Right→Wrong % |
> |---------------------|----------------|---------------|---------------|
> | 0.6200              | 39.8%          | 0.7%          | 11.7%         |
> | 0.6933              | 22.4%          | 1.0%          | 4.7%          |
> | 0.7200              | 9.7%           | 0.7%          | 1.7%          |
>
>
> With conservative thresholds, the Right→Wrong rate drops to nearly zero (0.0% for AMC, 1.7% for AIME), while Wrong→Right remains minimal but consistent (~0.7-1.0%). This demonstrates that our method primarily acts as an efficiency optimization rather than an accuracy enhancer. The small Wrong→Right improvements likely represent cases where early termination prevented overthinking, but this effect is marginal. Essentially, when thresholds are strict enough, we preserve the model's original problem-solving trajectory while achieving substantial token savings (9.7-16.1% even with conservative settings).
>
> ## Q4 (Weakness 2) Repetition-based Certainty Metric
>
> > Weakness 2: Certainty metric is primarily based on answer repetition: While effective, it may not capture deeper uncertainty or error modes beyond output consistency, especially when models are confidently wrong. The authors attempt to filter out hesitations (e.g., “wait”), but this is still heuristic.
>
> Surprisingly, early termination of a confidently wrong answer (request) is a feature, not a weakness when we apply Certaindex into real LLM inference engine. In the real-world deployment, a service provider wants to reduce the token expenditure to gain profit for completing a request correctly. If the model does not have the ability to answer the problem correctly, early terminating this request will help save tokens (thus costs). In our case, if a model makes a consistent answer to the question, even if it is wrong, it provides a strong signal that the model will not correct the answer in later iterations. Thus, early termination will benefit us from saving more tokens. Our method does not aim to change (increase) the model's accuracy - rather, we preserve the same accuracy distribution while achieving significant computational savings through early termination. The key value proposition is computational efficiency without sacrificing the model's inherent performance characteristics.
>
> Repetition-based certainty metric. We acknowledge this limitation of our repetition-based certainty metric. We note that our framework is flexible and allows for user-defined certainty metrics. In our experiments, we have also explored alternative approaches such as using reward model scores to define certainty (as shown in Section 3.1). This reward model-based approach can partially address the uncertainty and error modes beyond output consistency.

---

> > ### Comment · Reviewer_nxvt · 2025-08-06
> >
> > I thank the authors for their detailed response. My concerns are mostly addressed.

---

### Official Review · Reviewer_pntd · 2025-06-27

**Clarity:** 3
**Significance:** 2
**Originality:** 3
**Rating:** 4
**Confidence:** 3

**Summary:**

This paper proposes using Certaindex to address the issue of redundant tokens in large model reasoning tasks. The motivation is clear, and both experimental and theoretical analyses demonstrate that Certaindex can effectively eliminate redundant reasoning branches. Moreover, the approach is supported by a well-developed reasoning framework, Dynasor, built on SGLang, which significantly improves the efficiency of online batch inference systems.

**Questions:**

1. While thresholding-based allocation is acceptable, could it become difficult to manage as different data distributions and reasoning flows vary across tasks? If so, how is this issue addressed in the current design?

2. The “Probe-In-The-Middle” analysis in Figure 2 is insightful, but the visualization itself is not very intuitive. Is there a more straightforward or visually clear way to present this idea?

3. As mentioned in the weaknesses, can Certaindex reliably handle all types of reasoning paths when judging confidence? Is there a risk that it might interfere with certain reasoning strategies that the model would otherwise naturally follow?

**Ethical Concerns:**

["NO or VERY MINOR ethics concerns only"]

**Limitations:**

yes

**Quality:**

3

**Strengths And Weaknesses:**

**Strengths:**

1. The paper introduces Certaindex as a novel metric to assess the stability of intermediate reasoning outputs, aiming to determine whether the model has already converged on a correct answer. This is a timely and meaningful contribution, as reducing token usage during inference without compromising answer quality is a pressing challenge in LLM deployment.

2. The authors provide a complete and reusable implementation of their method through the Dynasor framework, built on SGlang. This not only validates the practicality of their approach but also offers valuable tooling for the research community, promoting reproducibility and further exploration.

3. The paper presents a well-rounded analysis, combining theoretical insights with empirical evaluations across multiple reasoning paradigms (e.g., Chain-of-Thought, Self-Consistency, MCTS, and REBASE). This strengthens the case for Certaindex as a general-purpose signal for early stopping in LLM reasoning.

**Weaknesses:**

1. While the method effectively reduces redundant tokens in high-confidence scenarios, it may also prematurely terminate reasoning when the model has not yet arrived at a correct answer. This could limit the model’s opportunity for self-correction and increase the risk of incorrect outputs. The paper would benefit from a more detailed discussion of such failure cases and how Certaindex behaves under different reasoning dynamics.

2. The evaluation is primarily focused on logic-intensive tasks such as mathematical reasoning, where the model’s confidence is easier to quantify. However, it remains unclear how well Certaindex generalizes to broader domains, such as open-ended or commonsense questions, where the model’s internal certainty may not align with actual correctness.

---

> ### Author Rebuttal · Authors · 2025-07-29
>
> Respond to pntd
>
> We thank Reviewer pntd for the thoughtful review and the positive feedback on Certaindex, Dynasor, and the breadth of our evaluation. We address the concerns and questions raised below.
> # Weaknesses
> ## W1: Risk of premature termination and failure cases.
> We acknowledged in the paper that early stopping introduces an accuracy-latency trade-off (Figure 8 row 3): if the threshold is set too aggressively, the model may terminate reasoning before self-correcting. Our theoretical analysis (Section 3.4 & Appendix E) specifically addresses this by providing a theoretical framework with a bound on the minimum number of reasoning steps required to preserve correctness with high probability. Moreover, our ablation on different thresholding methods (Figures 9 - 10) empirically explores this trade-off, demonstrating how different thresholds affect both accuracy and token savings. We will add an explicit discussion of failure cases in the revision, highlighting workload-specific behaviors and practical tuning strategies.
>
> ## W2: Generalization beyond math/logical reasoning.
> We acknowledge that our evaluation does not cover open-ended or commonsense questions. Our experiments focus on math and logic-heavy benchmarks because these tasks provide objective and easily verifiable correctness signals, which are critical for validating Certaindex. That said, Certaindex is inherently domain-agnostic, as it measures the convergence of reasoning patterns rather than relying on task-specific heuristics.
> While we have not included experiments on open-ended tasks, Dynasor’s framework can be readily extended to such domains, for example by incorporating an LLM-as-a-judge signal or other external rewards to approximate correctness. We will clarify this in the revision and discuss how threshold tuning can adapt to different workloads.
>
>
> # Questions
> ## Q1: Managing thresholds across tasks.
> It is true that the optimal threshold varies with data distribution and reasoning dynamics. Dynasor is designed for this flexibility: in practice, a service provider can collect statistics on model outputs for a given domain and tune thresholds using a small calibration set (similar to how temperature scaling is used for confidence calibration). We will clarify this point in the revision.
> ## Q2: Visualization of “Probe-In-The-Middle” (Figure 2).
> We appreciate the suggestion. While we aimed for completeness, we agree that the current visualization may appear dense. In a revised version, we will include a simplified version of the plot that highlights key trends (e.g., stability convergence) more clearly.
> ## Q3: Applicability to all reasoning paths.
> Certaindex is not tied to a single reasoning strategy. By tracking stability signals across any sequence of intermediate steps (e.g., CoT, SC, MCTS), it acts as a lightweight meta-controller rather than interfering with reasoning. Our evaluations (Sections 4) show that Certaindex complements existing reasoning flows without reducing their inherent flexibility.

---

> > ### Comment · Reviewer_pntd · 2025-08-07
> >
> > Thanks to the author, most of my concerns have been resolved.

---

### Official Review · Reviewer_Jsvc · 2025-07-01

**Clarity:** 3
**Significance:** 2
**Originality:** 2
**Rating:** 4
**Confidence:** 3

**Summary:**

This paper proposes a solution to solve the trade-off between increasing the token budget for sampling long chain of thoughts to improve accuracy vs. the compute inefficiency of overly long thoughts. They propose using uncertainty estimate based on the model's several intermediate answers in the long CoT to determine an early exit criteria, with the uncertainty calculated using discrete answer frequencies, or incorporating external reward models depending on the task. The paper then pairs this an adaptive scheduling algorithm (Dynasor) that allocates the token budget for batch and online workload. Empirical results show the effectiveness of Certaindex to estimate the correctness or number of steps before termination as well as the prowess/efficiency of their scheduling algorithm.

**Questions:**

- The "settling concept" mentioned in Line 31 has also been described in prior works: https://arxiv.org/abs/2504.07128
- Sec 2 para 2, the "probe-in-the-middle" is a standard test-time scaling approach as used in the S1 paper and should be cited as such.
- Certaindex using answer frequency to measure entropy for code, how does it compare to CodeT, another cross-consistency approach with unit test outputs and other methods relying on unit test feedback https://arxiv.org/abs/2207.10397?

**Ethical Concerns:**

["NO or VERY MINOR ethics concerns only"]

**Final Justification:**

I am satisfied with the effort authors have put into updating the related works which was my biggest issue. I think the papers contribution is reasonable.

**Limitations:**

Yes.

**Paper Formatting Concerns:**

Increase the space between figure captions and the remaining comments. E.g. end of Fig 4, Fig 5, top of Fig 2

**Quality:**

2

**Strengths And Weaknesses:**

**Strengths**
- The paper is well-written, easy to follow and addresses a useful problem addressing balancing the length of CoT and its inefficiencies.
- Convincing empirical results that demonstrate the effectiveness of Dynasor (the scheduling algorithm) over other baselines for batch and online workloads, demonstrating its utility to the community at large.

**Weaknesses**
- Certainty computation itself is not novel and quite similar to ideas used in prior works such as adaptive self-consistency (https://arxiv.org/abs/2305.11860). The adaptive scheduler basically determines the token budget in blocks/steps which is similar to automatically selecting the value of k (number of samples) and early stopping of the sampling process in adaptive SC. The authors should compare with this baseline in addition to self-consistency.
- Missing related works on using confidence estimates for inference-time decisions:
    - https://arxiv.org/abs/2305.11860
    - https://arxiv.org/abs/2311.09553
    - https://arxiv.org/abs/2411.04109
    - https://arxiv.org/abs/2504.05419, https://arxiv.org/abs/2504.15895 (concurrent)

---

> ### Author Rebuttal · Authors · 2025-07-29
>
> Respond to Jsvc
>
> We appreciate your time and suggestions in reviewing our work.  We take this opportunity to clarify our key contributions and address the concerns raised.
>
> # Weakness
>
> ## W1(a): Novelty of certainty computation.
> The reviewer suggests that *“certainty computation itself is not novel … such as adaptive self-consistency (Aggarwal et al., 2023).”* We stress that our contributions extend **far beyond** a single heuristic of certainty computation: we present a generalized metric (Certaindex), theoretical foundation, and system-level scheduler (Dynasor), making Dynasor both broadly applicable and practically efficient in serving real-world reasoning programs. Below, we clarify these contributions in detail.
>
> **(Generalizability of Certaindex.)**
> Unlike adaptive SC (Aggarwal et al.), Certaindex is a unified certainty metric that adapts to multiple reasoning paradigms beyond SC alone – including CoT, Monte Carlo Tree Search (MCTS), and Rebase – across a variety of workloads and models. Beyond introducing the metric, we conduct systematic ablation studies (Section 4.3) on different thresholding strategies for early stopping (Figures 9, 10), showing how these criteria affect both accuracy and token usage.
>
>
> **(Theoretical contribution.)**
> Unlike other works that rely on heuristics to perform early-stopping, we developed a theoretical framework (Section 3.4 & Appendix E) that provides provable guarantees for early exit and token allocation. This theoretical underpinning ensures that our certainty-based stopping criteria can be tuned for optimal trade-offs between performance and efficiency.
>
> **(System contribution.)**
> Our work is the first to apply Certaindex to build an end-to-end system (Dynasor) designed for optimizing token efficiency for LLM reasoning programs and reasoning models. Dynasor uses a system-level scheduler to optimize batch and online workloads with parallel decoding and dynamic token budgets allocation. This system design leads to substantial end-to-end latency and throughput improvements that prior approaches, such as Aggarwal et al., cannot achieve due to their sequential nature.
>
> Taken together, the generalizability of Certaindex, its theoretical guarantees, and its integration into a high-performance scheduler establish Dynasor as a comprehensive solution in efficiently scaling LLM reasoning, not just a repackaging of existing heuristics.
>
>
> ## W1(b): Comparison to https://arxiv.org/abs/2305.11860 (Aggarwal et al.) .
>
> Besides previously mentioned, we emphasize that an empirical comparison with adaptive SC (Aggarwal et al., 2023) is not meaningful, given the inherent differences in their underlying execution paradigms.
>
> **(1) Sequential vs. Parallel Decoding in real-world deployment.**
> Aggarwal et al.’s adaptive SC decodes one trajectory at a time, checking the stopping criterion after each generation. This sequential decoding results in latency proportional to the number of rounds × average tokens per round. In contrast, Dynasor exploits parallel decoding and dynamic token allocation, which significantly reduces latency in both batch and online workloads.
>
> **(2) Batch vs. Online Workload Performance.**
> While Aggarwal et al. can in theory reduce token counts in offline settings, it is not designed to optimize real-time throughput or latency. Dynasor’s scheduling explicitly addresses both token efficiency and online responsiveness, offering a better tradeoff between speed and accuracy.
>
> **(3) Conceptual Superset.**
> Our framework subsumes adaptive SC as a special case – in fact, an adaptive SC policy can be expressed as a particular instantiation of Dynasor with per-step thresholds. We provide a theoretical analysis of stopping criteria (Section 3.4) and explore multiple adaptive strategies (Figures 9–10), which covers the design space that adaptive SC inhabits.
>
> For these reasons, we believe running experiments against adaptive SC would not provide additional insight and could be misleading in evaluating online systems. We will clarify the relationship in the revision and explain why Dynasor generalizes and improves upon adaptive SC.
>
> ## W2: Related works
>
> We will cite these works in the final version of the paper.
>
> https://arxiv.org/abs/2305.11860 - *included in discussion above, omit here due to limited input length*
>
> https://arxiv.org/abs/2311.09553 - Kabra et al. focus on Program-Aided reasoning: they invoke external tools during inference and measure how well the combined LLM-plus-tool pipeline is calibrated. Their contribution is a calibration study and metric suite; they do not define an early-exit strategy, a token-budgeting algorithm, or any scheduling mechanism.
>
> https://arxiv.org/abs/2411.04109  - Prasad et al. present SCPO, which leverages SC during training to generate preference data for RLHF. All compute happens offline instead of online.
>
> https://arxiv.org/abs/2504.05419 - Zhang et al. (concurrent) learns a white-box linear probe on hidden states to predict correctness and exit early. Their method requires fine-tune the model and access to activations, and evaluated only on CoT.
>
> https://arxiv.org/abs/2504.15895 - Yang et al. (concurrent) propose a heuristic early-exit rule that monitors “wait” tokens in CoT and stops once a confidence threshold is met. The work provides neither theoretical guarantees nor a scheduler, and only target at CoT.
>
>
> # Questions
>
> ## Q1: “Settling concept” is introduced in https://arxiv.org/abs/2504.07128. (Marjanović et al.)
>
> We clarify that our contributions differ substantially from Marjanović et al.
>
> **Scope and Novelty**: Our contributions differ substantially from Marjanović et al. Their work is mainly a survey and analysis of Deepseek-R1 reasoning models, introducing a settling heuristic limited to that scope. In contrast, we show that this heuristic broadly exists in both reasoning models and non-reasoning models with reasoning programs (Section 3.2, Appendix C). We go further by developing a theoretical framework (Section 3.4, Appendix E) that rigorously bounds early exit without sacrificing accuracy – something not addressed by Marjanović et al. Beyond theory, our system-level contribution, Dynasor, extends this principle to real-world serving engines and delivers substantial efficiency gains on diverse workloads (GSM8K, AIME, LiveCodeBench), for both batch and online settings. In this sense, our work has a broader scope, a rigorous theoretical grounding, and system-level impact that clearly sets our work apart from Marjanović et al.
>
>
> **Timing and originality**: The work is arxiv-ed in May 2025, so the reviewer is correct that this work is concurrent to our work.
>
> ## Q2: Comparing with S1 (https://arxiv.org/abs/2501.19393, Muennighoff et al.)
>
> We respectfully disagree with the reviewer’s opinion.
>
> In fact, reviewer's comment overlooks a critical distinction: **S1 requires fine-tuning** the model to obtain the ability to scaling, whereas **Dynasor works out-of-the-box** with any model and reasoning programs by exploiting intrinsic signals during the reasoning process. This makes Dynasor far more practical and widely applicable across models and workloads, avoiding the costly retraining step required by S1. Moreover, S1 does not address a central question that Dynasor tackles: when should the model stop thinking without sacrificing accuracy? This is a key contribution of our work, supported by both a theoretical framework and extensive empirical results.
>
> We also disagree with the claim that “prompt in the middle” is a standard test-time scaling approach. To our knowledge, no closed-source models (e.g., OpenAI, Claude, Gemini) have publicly documented using this technique as a standard practice. We have also not found published works other than S1 that generalize this approach for test-time scaling without the need to fine-tune the model. Also note that this technique is limited to reasoning models and does not generalize to reasoning programs (e.g., Self-Consistency, Rebase, MCTS), which Dynasor supports but S1 does not.
>
> Taken together, our work does not require fine-tuning the model, provide theoretical grounding of the stopping criterion to save tokens without losing accuracy, and the generalizes to reasoning models and reasoning programs. Therefore, our contributions are substantially distinct from S1, both in goals and in method.
>
>
> ## Q3: Comparison with CodeT (https://arxiv.org/abs/2207.10397)
>
> CodeT and our work address different goals and settings.
>
> **Summary of CodeT**: CodeT automatically generates test cases to verify solutions for coding tasks, checking (1) the consistency of outputs against these tests and (2) agreement with prior code samples.
>
> **Key differences**:
> CodeT’s “cross-consistency with unit test feedback” fundamentally differs from Certaindex. CodeT is limited to coding tasks with strong external verifiers, whereas Certaindex measures certainty through answer frequency and entropy in a purely black-box LLM setting. In this sense, Certaindex is applicable to both coding and non-coding reasoning tasks, making it more generalized than CodeT.
>
> Moreover, CodeT does not address inference-time efficiency. Its evaluation (Section 4.3) focuses on the tradeoff between sample count vs accuracy, but not on token budgets or latency vs accuracy. In contrast, our work targets inference-time efficiency, aiming to minimize token usage while preserving accuracy across arbitrary inference-time algorithms, which is a more relevant problem in real deployment setup (i.e. price-per-query that the user needs to pay for).
>
> While external rewards or verifiers are beyond this paper’s scope, Dynasor’s flexible interface (Section 3.3 & Appendix D.1) can incorporate such signals. CodeT’s approach could be implemented as a specific policy within Dynasor, but our focus is on generality and efficiency without relying on test execution. In that sense, we don’t think comparing with CodeT is necessary.

---

> > ### Comment · Reviewer_Jsvc · 2025-08-03
> >
> > While I agree that the paper proposes a more general method and also provides theoretical justifications, my primary issue with the work is the lack of meaningful engagement with related work. I will agree with the authors that some of the points they make to contrast with existing work are fair and valid arguments, but are *nevertheless* needed to improve the quality and utility of the paper for the research community at large.
> >
> > Further, I would argue that the paper clearly benefits from the so-called "reasoning length inefficiency" motivation in the introduction, abstract and even title -- therefore, it is reasonable to expect a meaningful comparison with existing literature in this field acknowledging the basic similarities (even if not intentional) with other works. To this end, distinctions like offline vs online interventions while practically are meaningful, are not theoretically so far apart to not warrant a discussion at least in related work (while I agree need not be compared with).
> >
> > I acknowledge the arguments made by the authors, but unless I see a keen, good-faith interest from the authors to do a more in-depth literature survey with existing works and contrast, I would maintain my current score. While I understand the fervor of the authors in justifying their work, their very limited related works section (L351-363) in the main paper and lack of any such discussion in the appendix, gives me the impression that the authors are not keen to add it to the paper. On the other hand, I believe it will enhance the quality of this work and if so, I would be happy to increase my score to reward authors not overlooking similar or related work in this widely-studied field.

---

> ### Author Response · Authors · 2025-08-04
>
> We appreciate the reviewer’s feedback and fully agree that deeper engagement with related literature will enhance the quality and utility of the paper for the broader research community.
>
> Given the reviewer’s emphasis on this point, we now post an updated version of related work. We hope this is a more comprehensive review of prior work, particularly those sharing the motivation of reasoning programs, reasoning efficiency, even when differing in aspects such as intervention timing (offline vs. online) or theoretical framing.
>
> This expanded discussion will be integrated into a revised version of the paper, with full citations and careful contrasts. We hope this demonstrates our good-faith commitment to engaging deeply with the relevant literature, and we sincerely thank the reviewer for highlighting this important direction.
>
> **LLM Reasoning With Test-Time Scaling**. Large language models (LLMs) are increasingly being augmented with explicit step-by-step reasoning programs to tackle complex tasks. Chain-of-Thought (CoT) prompting [1] elicits step-by-step natural-language rationales, raising task accuracy with simple trigger phrases. Self-Consistency [2] enhances CoT by sampling multiple reasoning paths and selecting the most frequent answer, while Best-of-N sampling [3,4] generates multiple candidates and selects the highest-scoring solution. Tree-of-Thoughts (ToT) [5] casts reasoning as tree search, where nodes represent intermediate states and search algorithms explore promising branches, enabling backtracking and global evaluation. The ReAct [6] framework structures timesteps as (Thought, Action, Observation) triplets, allowing LLMs to reason internally and invoke external tools dynamically. More sophisticated approaches leverage Monte Carlo Tree Search (MCTS) [8,9], guided beam search[9], and REBASE [10] to systematically explore reasoning spaces. More recent work has established Test-Time-Scaling as a key paradigm for eliciting deeper reasoning from fixed-parameter LLMs without retraining [11]. The s1 framework introduces budget-forcing rules by appending "Wait" tokens or terminating early to dynamically manage compute allocation [12]. Further work shows that compute-optimal allocation favors parallel sampling over sequential deepening [13], that methods like SETS interleave sampling with self-verification [14], and that advances in meta-generation algorithms [15] and generative reward models [16] can further enhance inference-time scaling. These methods exploit multi-step reasoning, representing a fundamental shift toward inference-time scaling in modern LLM applications. However, the gains in accuracy often come at the cost of efficiency: multi-step reasoning produces far more tokens, leading to higher computational cost and latency. This trade-off sets the stage for Dynasor, which is a system designed to address these efficiency challenges by intelligently managing token usage and early exits.
>
> **Efficient CoT Reasoning**. Efficient LLM CoT reasoning targets at reducing excessive token usage in long reasoning chains [24]. Model merging blends fast shallow and thorough deep reasoning models for CoT inference [17,26]. Mid-generation self-evaluation prunes unpromising continuations on the fly [18]. Specialized fine-tuning compresses rationales by skipping irrelevant tokens [19], leveraging concise self-generated chains [20], applying length-aware tuning with O1-Pruner [21], and using reinforcement learning to trim overlong thought sequences [25,27]. Parameter-space controls such as CoT-Valve adjust chain length based on learned parameter cues [22]. Token-budget-aware frameworks estimate question difficulty to allocate computation dynamically [23]. FlashThink uses a verification model to identify when the model can stop reasoning without changing final output [29]. While each approach improves the trade-off between compute cost and reasoning quality, most depend on weight modifications, additional components, or extensive retraining—complicating real-world deployment. By contrast, our method adds only a lightweight proxy scheduler that manages reasoning depth within standard inference pipelines, preserving both simplicity and compatibility. We leverage certaindex as an indicator for more efficient CoT reasoning, consistent with recent findings that CoT reasoning often settles early [28].

---

> > ### Comment · Reviewer_Jsvc · 2025-08-05
> >
> > Thanks, I have raised my scores accordingly.

---

> ### Author Response · Authors · 2025-08-04
>
> **Certainty-Driven Early Stopping for Efficient LLM Reasoning**. Recent work has focused on cutting reasoning costs by using certainty-based signals. Adaptive-Consistency [30] reduces sampling by applying a lightweight, model-agnostic stopping rule. Early-Stopping Self-Consistency [31] lowers the cost of self-consistency by considering the answer consistency without hurting performance. Reasoning-Aware Self-Consistency [32] adjusts how many samples are needed based on the quality and consistency of reasoning paths. Besides, Program-aided reasoners's confidence more accurately reflects its correctness compared with pure CoT reasonings [45]. Concurrent work Dynamic Early Exit [33] detects high-confidence points during Chain-of-Thought reasoning to stop early. Self-Consistency Preference Optimization [34] improves reasoning by training the model to favor consistent answers. ConCISE [35] shortens reasoning steps by reinforcing token-level confidence and reducing unnecessary reflections. An information-theoretic method [36] introduces two metrics to measure how far the model’s reasoning is from ideal and how much each step contributes, stopping once confidence (measured via entropy) is high enough. Certainty-based Adaptive Reasoning [38] switches between short answers and long-form reasoning based on the model’s perplexity. Another method [37] checks hidden states to predict whether the answer is correct. However, these methods focus primarily on single reasoning paradigms (self-consistency or CoT), neglecting other reasoning algorithms and parallelism opportunities. In contrast, Certaindex provides a theoretically grounded stopping criterion that generalizes beyond self-consistency or CoT to multiple reasoning programs, including CoT with majority voting, MCTS, and reward-guided tree search, while adding only a lightweight proxy scheduler that preserves both simplicity and deployment compatibility.
>
> **Tool Validation Augmented Reasoning**. Several approaches have been proposed to enhance LLM generation by injecting external validation via test execution or unit‑test generation to improve answer quality. CodeT automatically generates unit tests with the same LLM, then applies dual execution consensus among code candidates to select the most reliable solution [39]. Self-Debugging prompts the model to detect when generated code fails, let it rubber‑duck the error, and autonomously patch the program based on runtime feedback [40]. LEVER trains a verifier that jointly considers the input task, code, and its execution outcomes to rerank and filter generated programs more accurately [41]. DOCE combines sample reranking, execution‑based scoring, minimum‑Bayes‑risk decoding, and self‑debugging into a single pipeline for execution‑aware decoding optimization [42]. While these methods rely on external tools, such as runtime environments, test runners, or auxiliary verifiers. Dynasor takes a different path: it only leverages the LLM itself, harnessing internal uncertainty and cross‑consistency signals to drive early stopping, without executing any code or requiring additional tooling.
>
> **LLM Inference Serving Systems**. Our contributions differ from general LLM serving frameworks. Systems like ParrotServe [43] and SGLang [44] optimize multi-query serving through smart scheduling and caching. ParrotServe co-designs frontend and backend with gang scheduling for dependent requests, while SGLang introduces structured primitives and maximizes KV cache reuse. These systems substantially improve throughput and latency for traditional LLM applications. However, they do not target LLM reasoning algorithms, which exhibit unique scaling behavior and within-request branching. Dynasor is the first system to exploit adaptive compute–accuracy trade-offs of advanced reasoning programs through a Certaindex-guided scheduler that dynamically allocates compute based on query difficulty.

---

> ### Author Response · Authors · 2025-08-04
>
> [1] Wei, Jason, et al. "Chain-of-thought prompting elicits reasoning in large language models." Advances in neural information processing systems 35 (2022): 24824-24837.
>
> [2] Wang, Xuezhi, et al. "Self-consistency improves chain of thought reasoning in language models." arXiv preprint arXiv:2203.11171 (2022).
>
> [3] Brown, Bradley, et al. "Large language monkeys: Scaling inference compute with repeated sampling." arXiv preprint arXiv:2407.21787 (2024).
>
> [4] Irvine, Robert, et al. "Rewarding chatbots for real-world engagement with millions of users." arXiv preprint arXiv:2303.06135 (2023).
>
> [5] Yao, Shunyu, et al. "Tree of thoughts: Deliberate problem solving with large language models." Advances in neural information processing systems 36 (2023): 11809-11822.
>
> [6] Yao, Shunyu, et al. "React: Synergizing reasoning and acting in language models." International Conference on Learning Representations (ICLR). 2023.
>
> [7] Feng, Xidong, et al. "Alphazero-like tree-search can guide large language model decoding and training." arXiv preprint arXiv:2309.17179 (2023).
>
> [8] Hao, Shibo, et al. "Reasoning with language model is planning with world model." arXiv preprint arXiv:2305.14992 (2023).
>
> [9] Xie, Yuxi, et al. "Self-evaluation guided beam search for reasoning." Advances in Neural Information Processing Systems 36 (2023): 41618-41650.
>
> [10] Wu, Yangzhen, et al. "Inference scaling laws: An empirical analysis of compute-optimal inference for problem-solving with language models." arXiv preprint arXiv:2408.00724 (2024).
>
> [11] Snell, Charlie, et al. "Scaling llm test-time compute optimally can be more effective than scaling model parameters." arXiv preprint arXiv:2408.03314 (2024).
>
> [12] Muennighoff, Niklas, et al. "s1: Simple test-time scaling." arXiv preprint arXiv:2501.19393 (2025).
>
> [13] Zhang, Qiyuan, et al. "What, how, where, and how well? a survey on test-time scaling in large language models." CoRR (2025).
>
> [14] Chen, Jiefeng, et al. "Sets: Leveraging self-verification and self-correction for improved test-time scaling." arXiv preprint arXiv:2501.19306 (2025).
>
> [15] Welleck, Sean, et al. "From decoding to meta-generation: Inference-time algorithms for large language models." arXiv preprint arXiv:2406.16838 (2024).
>
> [16] Mahan, Dakota, et al. "Generative reward models." arXiv preprint arXiv:2410.12832 (2024).
>
> [17] Wu, Han, et al. "Unlocking efficient long-to-short llm reasoning with model merging." arXiv preprint arXiv:2503.20641 (2025).
>
> [18] Manvi, Rohin, Anikait Singh, and Stefano Ermon. "Adaptive inference-time compute: Llms can predict if they can do better, even mid-generation." arXiv preprint arXiv:2410.02725 (2024).
>
> [19] Xia, Heming, et al. "Tokenskip: Controllable chain-of-thought compression in llms." arXiv preprint arXiv:2502.12067 (2025).
>
> [20] Munkhbat, Tergel, et al. "Self-training elicits concise reasoning in large language models." arXiv preprint arXiv:2502.20122 (2025).
>
> [21] Luo, Haotian, et al. "O1-pruner: Length-harmonizing fine-tuning for o1-like reasoning pruning." arXiv preprint arXiv:2501.12570 (2025).
>
> [22] Ma, Xinyin, et al. "Cot-valve: Length-compressible chain-of-thought tuning." arXiv preprint arXiv:2502.09601 (2025).
>
> [23] Han, Tingxu, et al. "Token-budget-aware llm reasoning." arXiv preprint arXiv:2412.18547 (2024).
>
> [24] Sui, Yang, et al. "Stop overthinking: A survey on efficient reasoning for large language models." arXiv preprint arXiv:2503.16419 (2025).
>
> [25] Hou, Bairu, et al. "Thinkprune: Pruning long chain-of-thought of llms via reinforcement learning." arXiv preprint arXiv:2504.01296 (2025).
>
> [26] Team, Kimi, et al. "Kimi k1. 5: Scaling reinforcement learning with llms." arXiv preprint arXiv:2501.12599 (2025).
>
> [27] Chen, Xingyu, et al. "Do not think that much for 2+ 3=? on the overthinking of o1-like llms." arXiv preprint arXiv:2412.21187 (2024).
>
> [28] Marjanović, Sara Vera, et al. "DeepSeek-R1 Thoughtology: Let's think about LLM Reasoning." arXiv preprint arXiv:2504.07128 (2025).
>
> [29] Jiang, Guochao, et al. "Flashthink: An early exit method for efficient reasoning." arXiv preprint arXiv:2505.13949 (2025).
>
> [30] Aggarwal, Pranjal, Aman Madaan, and Yiming Yang. "Let's Sample Step by Step: Adaptive-Consistency for Efficient Reasoning and Coding with LLMs." arXiv preprint arXiv:2305.11860 (2023).
>
> [31] Li, Yiwei, et al. "Escape sky-high cost: Early-stopping self-consistency for multi-step reasoning." arXiv preprint arXiv:2401.10480 (2024).
>
> [32] Wan, Guangya, et al. "Reasoning aware self-consistency: Leveraging reasoning paths for efficient llm sampling." arXiv preprint arXiv:2408.17017 (2024).
>
> [33] Yang, Chenxu, et al. "Dynamic Early Exit in Reasoning Models." arXiv preprint arXiv:2504.15895 (2025).
>
> [34] Prasad, Archiki, et al. "Self-consistency preference optimization." arXiv preprint arXiv:2411.04109 (2024).
>
> [35] Qiao, Ziqing, et al. "ConCISE: Confidence-guided Compression in Step-by-step Efficient Reasoning." arXiv preprint arXiv:2505.04881 (2025).

---

> > ### Author Response · Authors · 2025-08-04
> >
> > [36] Yong, Xixian, et al. "Think or Not? Exploring Thinking Efficiency in Large Reasoning Models via an Information-Theoretic Lens." arXiv preprint arXiv:2505.18237 (2025).
> >
> > [37] Zhang, Anqi, et al. "Reasoning Models Know When They're Right: Probing Hidden States for Self-Verification." arXiv preprint arXiv:2504.05419 (2025).
> >
> > [38] Lu, Jinghui, et al. "Prolonged reasoning is not all you need: Certainty-based adaptive routing for efficient llm/mllm reasoning." arXiv preprint arXiv:2505.15154 (2025).
> >
> > [39] Chen, Bei, et al. "Codet: Code generation with generated tests." arXiv preprint arXiv:2207.10397 (2022).
> >
> > [40] Chen, Xinyun, et al. "Teaching large language models to self-debug." arXiv preprint arXiv:2304.05128 (2023).
> >
> > [41] Ni, Ansong, et al. "Lever: Learning to verify language-to-code generation with execution." International Conference on Machine Learning. PMLR, 2023.
> >
> > [42] Li, Haau-Sing, et al. "Doce: Finding the sweet spot for execution-based code generation." arXiv preprint arXiv:2408.13745 (2024).
> >
> > [43] Lin, Chaofan, et al. "Parrot: Efficient serving of {LLM-based} applications with semantic variable." 18th USENIX Symposium on Operating Systems Design and Implementation (OSDI 24). 2024.
> >
> > [44] Zheng, Lianmin, et al. "Efficiently Programming Large Language Models using SGLang." (2023).
> >
> > [45] Kabra, Anubha, et al. "Program-aided reasoners (better) know what they know." arXiv preprint arXiv:2311.09553 (2023).

---

### Decision · Program_Chairs · 2025-09-17

**Decision:**

Accept (poster)

**Comment:**

This paper addresses the problem of excessively long latent reasoning chains in LLMs prompted to answer problems requiring multi-step thinking. A heuristic method is proposed to detect whether reasoning can be terminated early by injecting a prompt that probes for a guessed answer at intermediate steps and tracking the consistency of these guesses. The method is integrated into a dynamic compute allocation framework, which delivers a reduction in total compute cost when amortised over multiple queries under a fixed budget. Experiments on maths benchmarks show that this approach does not harm accuracy while significantly reducing average response length.

Strengths of the paper:
- Good motivation for and presentation of the algorithm.
- Strong empirical results and a system that makes the method readily deployable.
- Thorough empirical evaluation and analysis.

Main weaknesses:
- Comparison with related work (Jsvc). In the rebuttal, the authors provided an extensive related work discussion, which satisfied the reviewer who raised this concern, and this should be included in the revision.
- Scope of the tasks evaluated on is somewhat narrow (pntd). The authors agreed that the method is developed with a focus on maths / "logic-heavy" tasks. The reviewer who raised this concern was satisfied with the authors' responses.
- Some of the design choices and motivations could be better justified and failure modes better analysed (pntd, nxvt). This was answered in the rebuttal with additional discussion and experiments, which should be included in the revision.

Overall, the paper is just above the bar for acceptance, and the authors are encouraged to include the additional discussion and experiments from the rebuttal in the final version.